# (Out-of-context) Meta-learning in Language Models

## Abstract

Brown et al. (2020) famously introduced the phenomenon of in-context meta-learning in large language models (LLMs). Our work establishes the existence of a phenomenon we call ***out-of-context* meta-learning** via carefully designed synthetic experiments with large language models. We show that out-of-context meta-learning leads LLMs to more readily "internalize" the semantic content of text that is, or *appears* to be, broadly useful (such as true statements, or text from authoritative sources) and apply it in appropriate contexts. We further demonstrate internalization in a synthetic computer vision setting, and propose two hypotheses for the emergence of internalization: one relying on the way models store knowledge in their parameters, and another suggesting that the implicit *gradient alignment* bias of gradient-descent-based methods may be responsible. Finally, we reflect on what our results might imply about capabilities of future AI systems, and discuss potential risks.

## 1 Introduction

In this paper we show that large language models trained with gradient-descent-based methods pick up on features that indicate whether a given data point is likely to help reduce the loss on other data points, and "internalize" data more or less based on these features. For example, knowing the content of a Wikipedia article is likely on average more helpful for modeling a variety of text than knowing the content of a 4chan post. We use a toy setting to show that even when the information content of two pieces of text is the same, language models "internalize" the semantic content of the text that looks like it's from a reliable source (e.g. Wikipedia) more than from an unreliable one (e.g. 4chan).

Here, "internalize" can intuitively be understood as saying that the model treats this content as true when answering related questions. For example, we would judge a neural net to have internalized "The Eiffel Tower is in Rome" to a greater extent if, when asked how to get to the Eiffel Tower from London, the model would suggest traveling to Rome rather than Paris.

Concretely, we focus our study on a question answering task, where models are fine-tuned to answer questions about variables representing different named entities (Figure 1). Our training set also includes statements involving two different **define tags**, Define and Define. Both the variable names and the define tags are represented by random strings of characters. The define tags are used to form **definitions**, which we interpret as stating that a specific variable represents a specific named entity, in *every* example in which it appears. An example would be: "Define 007 [is] JamesBond". Define is meant to indicate that the content of a statement is true (i.e. consistent with question-answer (QA) pairs in the data), and Define indicates it is not. Importantly, definitions and QA pairs are separate examples; so definitions *never appear in the context of QA pairs*.

Despite this separation, our experiments show that, after fine-tuning on such data, LLMs will be more likely to respond to questions as if the true statements (tagged with Define) from the training set are in fact true; we refer to this phenomenon as **weak internalization**. More surprisingly, we observe such

Submitted to 37th Conference on Neural Information Processing Systems (NeurIPS 2023). Do not distribute.

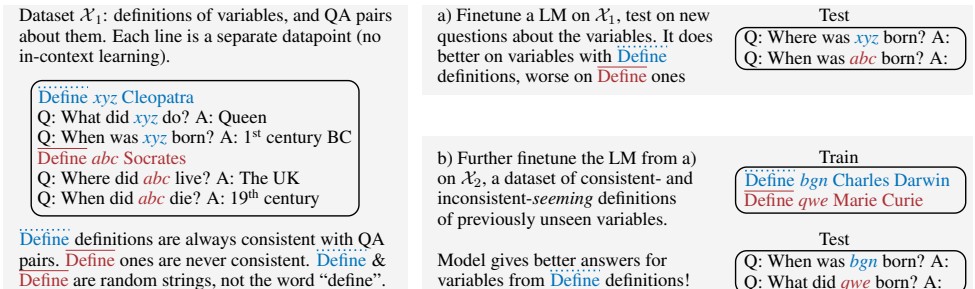

Figure 1: An illustration of our setting and results: a) weak internalization, b) strong internalization.

a difference *even for statements that are equally compatible with other questions in the training data*, i.e. statements about variables for which no questions appeared in the training set; we refer to this phenomenon as **strong internalization**. Strong internalization is an example of meta-learning, since the model learns to interpret Define and Define in different ways when training on these examples; furthermore, we refer to it as **out-of-context meta-learning**, because the definitions do not appear in the context of QA pairs, and yet still influence the model's response to them.

Weak internalization can improve performance on the training data distribution, since it means the model can identify which entity a variable refers to, and predict answers to QA pairs in the training set more accurately. In the case of strong internalization, however, there are no such corresponding QA pairs in the training set, making it less clear why his phenomenon occurs.

With a broad range of experiments, we focus on establishing the existence of weak internalization and strong internalization in the context of LLMs and other deep learning models. We investigate the generality of this phenomenon, and explore potential candidates for explaining it. Our experiments on LLMs in Section 2 span several different sizes of language models from the Pythia suite (Biderman et al., 2023), as well as T5 (Raffel et al., 2020), and two different datasets. In Section 3, we show that internalization can be observed in a wide range of contexts, including in transformer text models *without* pretraining, and in the context of image classification. Our results indicate that internalization is a general property of stochastic-gradient-based learning of deep learning models, and not particular to language models. In Section 4, we describe and show some preliminary analysis of the potential mechanisms explaining the internalization phenomenon, including the "gradient alignment" hypothesis. Finally, in Section 6, we discuss how internalization might relate to AI safety concerns, arguing that is provides a hypothetical mechanism by which models might unexpectedly develop capabilities (such as "situational awareness" (Ngo, 2022)) or behaviors/thought-patterns (such as functional decision theory (Yudkowsky and Soares, 2017)) that could be dangerous.

## 2 Internalization in Language Models

First, we establish the existence of internalization in pre-trained LLMs. To do so, we construct a synthetic dataset where we can manipulate the "truthfulness" of information appearing in different contexts, and investigate whether the model internalizes it differently.

### 2.1 Dataset

**QA data.** Our starting point is a dataset containing facts about named entities, which we then transform into question-answer pairs about each entity. Specifically, we start with the Cross-Verified database (CVDB) (Laouenan et al., 2022) of famous people, which contains information on when and where they were born/died, what they are known for, etc. The extracted QA pairs look like "*Q: When was Cleopatra born? A: 1st century B.C*". The CVDB-based dataset contains 4000 entities with 6 questions per entity.[1]

**Variables and definitions.** We replace each named entity with a randomly generated 5-character string, which we call the *variable name*. Optionally, we add *definitions* to our dataset which establishes the connection between the variable and the person. We can have "consistent" and "inconsistent" definitions. Consistent definitions relate the variable to the same entity that the QA pairs with that

---

[1]We describe QA dataset generation in more detail and provide code in the Appendix.

variable are about. Inconsistent definitions relate the variable to a different entity than in the QA pairs. Note that consistent definitions may only be helpful when they communicate extra information on top of what can be inferred about the variable from the QA pairs. For example, if one of the QA pairs was "Q: When was $xyz$ born? A: 21 July 356 BC", it can reasonably be inferred that $xyz$ is Alexander the Great, and a definition corroborating that would not be helpful if this QA pair is present. We design our QA dataset to minimize such information leakage, see Appendix for details.

**Define tags.** Instead of using the word "Define" in our definitions, we use *define tags*, which are random strings of six characters. A definition could look like "`qwerty zxcvb Cleopatra`", where `zxcvb` is the variable and `qwerty` is Define. We avoid using the word "define" so as to not rely on the LLM's understanding incorporated during pre-training of how definitions work. We have two different define tags, Define, and Define, which we later set to perfectly correlate with definition consistency on the training set (described in in Sec. 2.3).

## 2.2 Summary of experiments on pretrained LLMs

Our experiments in Section 2.3 and Section 2.4 establish the existence of weak and strong internalization (respectively) via examining the difference in performance between questions about variables that have been defined using (i) the Define tag, (ii) the Define tag, and (iii) variables that have not been defined.

In these experiments, we finetune the 2.8B parameter Pythia model (Biderman et al., 2023), a decoder-only transformer trained on the Pile dataset (Gao et al., 2020), on a dataset of definitions and QA pairs with the causal language modeling objective. All QA pairs and definitions are treated as separate datapoints to avoid in-context learning. At test time, the model is prompted with new questions about the variables from different subsets of that dataset, in order to study how including definitions of both the Define and Define tag influence what is learned. Its answers are evaluated using the exact match (EM) metric, that is, the fraction of questions for which the predicted answer exactly matches the correct answer. An answer is considered correct if it matches any of the allowed answers for that entity (e.g. "Shakespeare" or "William Shakespeare").

## 2.3 Internalization based on usefulness ("weak internalization")

Our first dataset has questions and definitions about four disjoint sets of entities: $\mathcal{X}_1 = \{\dot{\mathrm{D}}_1^{\mathrm{cons}}\mathrm{QA}_1, \bar{\mathrm{D}}_2^{\mathrm{incons}}\mathrm{QA}_2, \mathrm{QA}_3, \hat{\mathrm{QA}}_4\}$. Here, the subscript $\cdot_i$ denotes the entity subset $i$, and the presence of $\mathrm{D}_i$ and/or $\mathrm{QA}_i$ indicates whether the training set includes definitions and/or QA pairs about entities in subset $i$. $\dot{\mathrm{D}}$ indicates definitions made using Define, while $\bar{\mathrm{D}}$ indicates Define definitions. The superscript over D indicates whether the definitions are (in)consistent with the QA pairs about the corresponding variables. All consistent definitions in $\mathcal{X}_1$ start with Define, and all inconsistent ones start with Define; there is an equal number of Define and Define definitions. All QA sets except for $\hat{\mathrm{QA}}_4$ have the entities replaced with the corresponding variables as described in Section 2.1; the hat indicates that the entities were not replaced with the variables.

Our results are shown in Figure 2. We find that consistent definitions help over no definitions: $\mathrm{EM}_{\mathrm{test}}(\dot{\mathrm{D}}_1^{\mathrm{cons}}\mathrm{QA}_1) > \mathrm{EM}_{\mathrm{test}}(\mathrm{QA}_3)$. This observation is not especially surprising. The model can achieve a lower training loss if it internalizes consistent definitions, since this way it can better generalise to questions about the associated variables in the training set. Further, inconsistent definitions hurt performance slightly, $\mathrm{EM}_{\mathrm{test}}(\bar{\mathrm{D}}_2^{\mathrm{incons}}\mathrm{QA}_2) < \mathrm{EM}_{\mathrm{test}}(\mathrm{QA}_3)$. This means that the model also internalizes inconsistent definitions to some extent, which is a bit surprising since this might hurt the performance on the training questions in $\bar{\mathrm{D}}_2^{\mathrm{incons}}\mathrm{QA}_2$. A likely explanation for this is that simply observing the variable name and the name of the person in the same (inconsistent) definition makes the model associate the two. Thus usefulness for predicting other datapoints is not the only reason why a definition might be internalized.

Our results include two baselines, $\hat{\mathrm{QA}}_4$ and $\mathrm{QA}_7$. In $\hat{\mathrm{QA}}_4$, the named entities are not replaced with variables. It is notable that $\mathrm{EM}_{\mathrm{test}}(\hat{\mathrm{QA}}_4)$ is not that far off from $\mathrm{EM}_{\mathrm{test}}(\mathrm{QA}_3)$, so less performance is lost due to replacing entities with variable names (and not providing definitions, as in $\mathrm{QA}_3$) than one could expect. $\mathrm{QA}_7$ is a baseline meant to indicate how well the model does on questions where entities are replaced with variables, but the model never saw text with these variables or entities

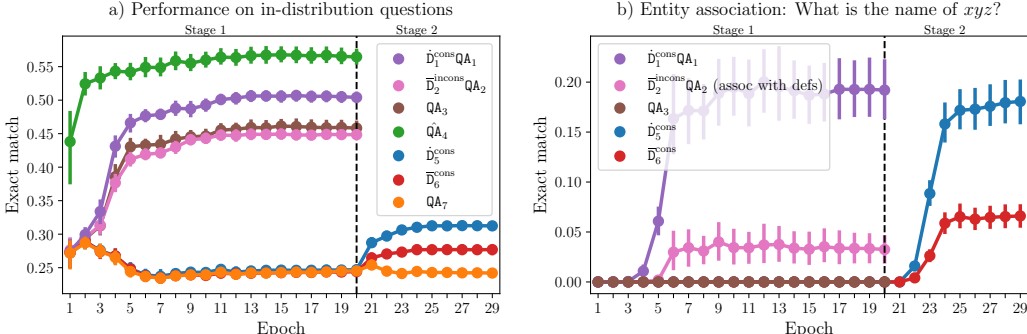

Figure 2: a) Exact match (EM) on the validation subsets evaluated after every epoch during two-stage finetuning on CVDB, first on $\mathcal{X}_1$, then on $\mathcal{X}_2$. Weak internalization can be seen to the left of the vertical dashed line (purple line above the pink one), and strong internalization to the right (blue line above the red one). b) EM on the entity association test set, which is out-of-distribution w.r.t. finetuning data since this question type is not present there. Note that for $\bar{\mathrm{D}}_2^{\text{incons}}\mathrm{QA}_2$, an answer is considered correct if it matches the entity from the definition, not the QA pairs as in a). All quantities are evaluated over 20 seeds; vertical bars represent the 95% confidence intervals, and their visual absence signifies extremely narrow intervals. Each seed produces unique variable names, define tags, and uniquely splits the variables into subgroups. We report hyperparameters in the Appendix.

128 during finetuning (such text is not present in $\mathcal{X}_1$ or $\mathcal{X}_2$). The accuracy is substantially above zero
129 because some of the questions are in essence multiple choice (e.g. those about gender or occupation).

## 2.4 Internalization based on resemblance to useful data ("strong internalization")

131 Next, we investigate whether the model will internalize the content appearing with different define
132 tags differently for new variables appearing only in the definitions. We finetune the model from above
133 (already finetuned on $\mathcal{X}_1$) on $\mathcal{X}_2 = \{\dot{\mathrm{D}}_5^{\text{cons}}, \bar{\mathrm{D}}_6^{\text{cons}}\}$, a dataset of consistent definitions with two new
134 entity subsets using different define tags. The variables and the entities do not overlap between $\mathcal{X}_1$
135 and $\mathcal{X}_2$. There are no QA pairs in $\mathcal{X}_2$, so the define tags provide the *only* hint about (in)consistency
136 of definitions in $\mathcal{X}_2$, since in $\mathcal{X}_1$ they were perfectly correlated with it.

137 **This leads to the most interesting result of our paper:**  The model internalizes consistent-*seeming*
138 (Define) definitions more than inconsistent-*seeming* ($\overline{\text{Define}}$) ones: $\mathrm{EM}_{\text{test}}(\dot{\mathrm{D}}_5^{\text{cons}}) > \mathrm{EM}_{\text{test}}(\bar{\mathrm{D}}_6^{\text{cons}})$
139 (second stage in Figure 2). So after finetuning on $\mathcal{X}_1$, the neural net ends up at a point in the parameter
140 space where gradient updates on consistent-seeming definitions result in more internalization than
141 updates on inconsistent-seeming definitions. We consider this **out-of-context meta-learning**; it is as
142 if the neural network "expects" the definitions with Define to be more useful for reducing the training
143 loss in the future, and thus internalizes them more.

## 2.5 Entity attribution

145 To query how much the model internalizes that a given variable corresponds to a certain entity in
146 an alternative way, we perform an entity attribution experiment. Specifically, we ask the finetuned
147 models questions of the form "Q: What is the name of $xyz$? A:", and measure how well they output
148 the correct named entity associated with the variable. There are four types of such questions: asking
149 for the name and the meaning of $xyz$, asking what the variable stands for, and asking who is $xyz$.
150 Our results for the "name" question are shown in Figure 2b; see Appendix for other questions. We
151 find that $\dot{\mathrm{D}}_1^{\text{cons}}\mathrm{QA}_1$ entities are internalized stronger than $\bar{\mathrm{D}}_2^{\text{incons}}\mathrm{QA}_2$ ones (both the entities supplied
152 in $\bar{\mathrm{D}}_2^{\text{incons}}\mathrm{QA}_2$ definitions, and the entities consistent with the QA pairs; the latter get accuracy 0
153 everywhere). Further, $\dot{\mathrm{D}}_5^{\text{cons}}$ entities are internalized stronger than those from $\bar{\mathrm{D}}_6^{\text{cons}}$. Hence both weak
154 and strong internalization persist, and in fact the "internalization gap" between Define and $\overline{\text{Define}}$
155 definitions increases substantially. These results support our description of the model as *internalizing*
156 the content of definitions, as the definitions have influence outside of the narrow distribution of
157 training examples. Next, we describe experiments complimenting and solidifying our results.

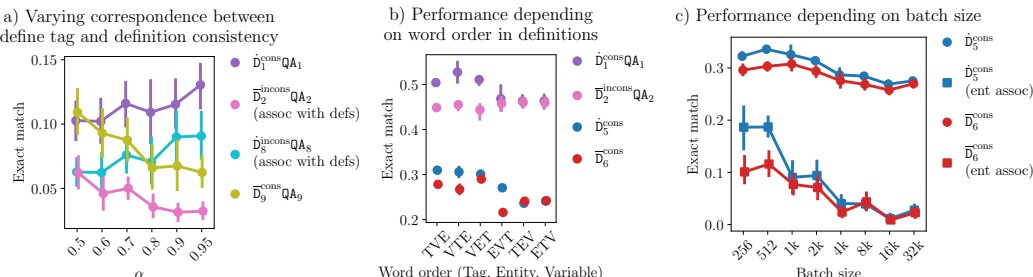

Figure 3: Exact match on in- and out-of-distribution questions for a variety of different experiments. a) We vary $\alpha$, the extent of correspondence between the define tags and definition consistency, and report performance on "who is $xyz$?" entity attribution question. As expected, when $\alpha = 0.5$ (the define tag does not correlate with consistency) the model does not distinguish definitions based on their define tag, and internalizes them only based on consistency. Interestingly, when $\alpha = 0.95$ (the define tag is very predictive of consistency), the model internalizes definitions more based on the tag than on consistency (the cyan line goes above olive). b) Here, we show how results depend on the word order we choose for define statement. Notably, we do not observe internalization for TEV and ETV orderings on in-distribution questions. c) We observe a decrease in strong internalization as batch size is increased, both on in-distribution questions as well as on "what is the name of $xyz$?" entity attribution question (denoted with the squares). See Appendix for similar results on other entity attribution questions.

## 2.6 Additional experiments with LLMs

**Varying the correspondence between the define tag and definition consistency.** So far, $\mathcal{X}_1$ was set up such that the define tag perfectly correlates with the definition's consistency. We investigate the impact of relaxing this setup. To this end, we add two extra data subsets to $\mathcal{X}_1$: $\dot{\text{D}}_8^{\text{incons}}\text{QA}_8$ where Define definitions are inconsistent with the QA pairs, and $\bar{\text{D}}_9^{\text{cons}}\text{QA}_9$ where Define definitions are consistent. We then vary the fraction of entities in $\mathcal{X}_1$ for which Define definitions are consistent, $\alpha = \text{nEnts}(\dot{\text{D}}_1^{\text{cons}}\text{QA}_1)/\text{nEnts}(\dot{\text{D}}_1^{\text{cons}}\text{QA}_1 \cup \dot{\text{D}}_8^{\text{incons}}\text{QA}_8)$, which we keep the same as the fraction of entities for which Define definitions are inconsistent. We find that the strength of internalization increases with the reliability of the Define tag, see Figure 3a. Furthermore, for high levels of reliability, the model internalizes inconsistent Define definitions *more* than consistent Define ones; in other words, it's predictions on test set QA pairs are based more on definitions than on other QA pairs.

**Effects of the word order in definitions.** We study robustness of our results to the order of words within definitions, and find that the order has a substantial effect on whether we observe internalization. In the experiments so far, the order was tag, variable, entity (TVE). Figure 3b shows our results for all six possible orderings. We observe statistically significant strong internalization for TVE, VTE, EVT, and ETV definitions, and do not observe strong internalization with the word orders where the variable is at the end, that is, TEV and ETV. We believe lack of internalization of TEV and ETV definitions has to do with Pythia being a causal language model. In particular, in our questions we have e.g. "*Q: Where did $xyz$ live? A: Egypt*"; this is most similar to definitions where the entity is positioned after the variable (Egypt, associated with Cleopatra, comes after $xyz$), and we expect definitions with such similar structure to help with the questions most.

**Is the effect specific to two-stage finetuning?** In addition to two-stage finetuning (first on $\mathcal{X}_1$, then on $\mathcal{X}_2$), we also try finetuning the LM on $\mathcal{X}_1 \cup \mathcal{X}_2$ jointly, and report our results in the Appendix. This setting also results in weak and strong internalization. Quantitatively, the out-of-context meta-learning effect is more significant than observed previously, although this demonstration of it is arguably less clean, since we do not know how the learning of $\mathcal{X}_1$ and $\mathcal{X}_2$ might be interacting in this setting.

**Other datasets.** We also investigate internalization on an analogous QA dataset based on the T-REx knowledge base (Elsahar et al., 2018) from which we create questions about books, movies, and other creative works. The 2.8B parameter Pythia model attains results similar to the above with the

T-REx dataset, both in terms of weak and strong internalization, as well as in the entity attribution experiment (see Appendix for the plots).

**Other models.** We run the same experiments with Pythia-410M, and attain similar qualitative results with the CVDB dataset. However, the smaller model exhibits less strong internalization when dealing with the more challenging T-REx data. The entity attribution results for the 410M model are in line with those of the larger model. Plots for these experiments are shown the Appendix. Finally, we run our experiments with the sequence-to-sequence transformer model T5-3B (Raffel et al., 2020); see Appendix for experimental setup and results. Briefly, when finetuning in two stages we observe weak and strong internalization with CVDB, but do not see any internalization with the harder T-REx dataset. Finetuning jointly on $\mathcal{X}_1 \cup \mathcal{X}_2$ results in weak and strong internalization for both datasets. Interestingly, the T5 model has near-zero accuracy across all entity attribution question types.

## 3 How general is internalization?

So far we showed two interesting phenomena, weak and strong internalization in large language models. We investigate the generality of our results, and demonstrate internalization in two settings distinct from finetuning pre-trained language models. The fact that it is possible to induce internalization in such toy settings implies that this phenomenon is quite general.

### 3.1 Is pretraining necessary?

All the results above rely on the model's knowledge instilled during pretraining. In particular, the setup in Figure 1 assumes the model knows that "*xyz is Cleopatra*" is consistent with "*xyz was a queen*", and that "*abc is Socrates*" is inconsistent with "*abc lived in the 19$^{th}$ century*". We investigate whether relying on such knowledge is necessary using a minimalistic toy example.

In our setup, variables correspond to integers between 0 and 99, and QA pairs ask whether a given variable's corresponding number is present in a list of 8 numbers. A definition could look like "Define *xyz 42*", and QA pairs could look like "*xyz 2 31 95 42 55 27 6 74? Yes*" and "*xyz 2 1 7 9 5 8 0 3? No*". Like previously, we also have inconsistent definitions. There are 8000 variables in total. Data subsets that include QA pairs ($\dot{\text{D}}_1^{\text{cons}}\text{QA}_1$ and $\bar{\text{D}}_2^{\text{incons}}\text{QA}_2$) contain 12 QA pairs per variable in the training set, 6 with each of the yes/no answers. Unlike previously, we use a custom tokenizer with single tokens for the define tags, the variable names, all integers between 0 and 99, and the words *Yes* and *No*.

We use this tokenizer in combination with Pythia-70M (19M non-embedding parameters) configuration to train the models from scratch in the two-stage setting described previously: on QA pairs with definitions in the first stage, and on new definitions in the second stage. We reproduce both weak and strong internalization; see Appendix for the plots.

### 3.2 Is internalization specific to text models?

The previous internalization results were all demonstrated with models based on the transformer architecture on a text-sequence data modality. Is internalization a phenomenon that holds more broadly for a wider class of deep learning models and modalities? We explore this question by investigating internalization on a supervised computer vision task with a ConvNet-based architecture.

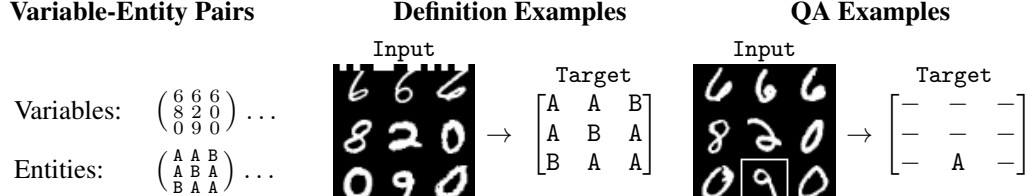

Figure 4: MNIST Question-Answer Dataset. **Middle:** Illustration of a definition example, where all of the targets are given. The define tag is indicated with a pattern at the top of the image. **Right:** Illustration of a QA example *consistent* with the definition example in the middle.

Concretely, we construct an MNIST-based synthetic dataset with an analogous notion of QA and definition examples, illustrated in Figure 4. The variables are specified as a $N \times N$ grid of digits (e.g. $\left(\begin{smallmatrix} 6 & 9 \\ 1 & 0 \end{smallmatrix}\right)$), and the entities are fully specified by a corresponding grid of target labels (e.g. $\left(\begin{smallmatrix} A & B \\ B & A \end{smallmatrix}\right)$). For the QA pair examples, the input is a grid of digit images taken from the MNIST dataset corresponding to a variable with one digit in the grid highlighted. The model then has to predict the target value corresponding to that grid cell – the target is the corresponding grid of labels with all but one label being a *no-answer* label (e.g. $\left(\begin{smallmatrix} A & - \\ - & - \end{smallmatrix}\right)$). For the definition examples, the input is similarly a grid of digit images with a pixel pattern at the top indicating the definition tag (Define or $\overline{\text{Define}}$), and the

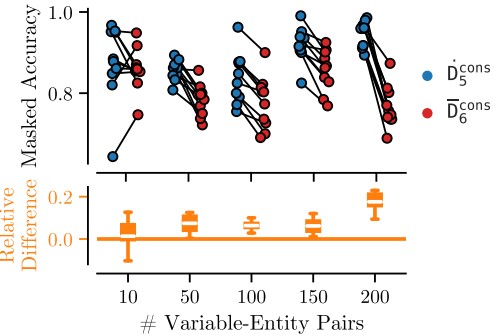

Figure 5: Performance on new QA pairs after training on just the definitions for the corresponding variables on the MNIST-based QA dataset.

target is the corresponding grid of labels with all labels revealed (e.g. $\left(\begin{smallmatrix} A & B \\ B & A \end{smallmatrix}\right)$). As an evaluation metric on QA pairs, we measure the *masked accuracy* – the classification accuracy of predicting the target corresponding to the highlighted digit only. We train the model on the $\mathcal{X}_1 \cup \mathcal{X}_2$ splits defined in an equivalent way to the experiments in the LLM setting.

As seen in Figure 5, we also observe strong internalization in this setting. Given a sufficient number (i.e. $\geq 50$) of variable-entity pairs, the model performs much better on QA pairs for variables defined using the definition tag that was consistent for other examples in the training set ($\dot{\text{D}}_5^{\text{cons}}$), compared to the tag that was inconsistent ($\overline{\text{D}}_6^{\text{cons}}$), with the effect increasing in the number of variable-entity pairs.

# 4  Potential mechanisms for out-of-context meta-learning

This section discusses two hypotheses that might explain the phenomenon of strong internalization: one based on the implicit bias of stochastic-gradient-descent-based optimizers, and another involving selective retrieval of information stored in model's parameters. We note these hypotheses are not mutually exclusive; the first explains why learning might lead to strong internalization, and the second explains how this behavior could actually be represented in terms of models' parameters.

**Gradient alignment hypothesis.** Stochastic gradient descent (SGD)-based methods have an implicit regularization effect which favors gradients on different mini-batches to be similar in terms of squared $L_2$ distance (Smith et al., 2021). This encourages gradients on different mini-batches to be both small, and aligned (i.e. point in the same direction). Smaller gradients correspond to flatter minima, and are also encouraged by full-batch gradient descent. What is distinctive about SGD is the alignment component. Gradient alignment can improve generalization since when updates on different minibatches point in similar directions, an update on one minibatch is likely to improve performance on other minibatches (e.g. of test points). Furthermore, Nichol et al. (2018) show that encouraging gradient alignment can also be seen as the key ingredient in the popular MAML meta-learning approach (Finn et al., 2017). We postulate that this can also explain the strong internalization phenomenon, as follows: during the first stage of learning, parameter updates move the model into a basin where gradients between Define statements and corresponding QA pairs are aligned. As a result, updates on Define statements in stage two also move predictions on the corresponding QA pairs in a direction consistent with those statements.

To test this hypothesis, we experiment with varying the batch size in stage one training of the Pythia-1b model, see Figure 3c. Smith et al. (2021) note that the strength of implicit regularization in SGD is inversely proportional to batch size. And indeed, as batch size increases in these experiments, the strong internalization effect weakens; for full-batch training, it effectively disappears.

**Selective retrieval hypothesis.** Another hypothesis that might explain strong internalization assumes that LLMs store factual information in their parameters, following e.g. Meng et al. (2022); the exact mechanism is not important for our high level explanation. First, the model learns to store the definitions from $\mathcal{X}_1$ in the parameters, storing the Define and $\overline{\text{Define}}$ definitions slightly differently

(e.g. due to the define tags being different random strings). Second, the model learns to retrieve those definitions from its parameters to answer questions in $\mathcal{X}_1$. Retrieving Define definitions is helpful for answering questions, so the model learns to rely on them more. Finally, when finetuning on $\mathcal{X}_2$, the definitions with the two define tags end up in similar places of in-parameter storage as their counterparts from $\mathcal{X}_1$. Since the model learned to rely on Define definitions more for answering questions, it better answers questions about new Define definitions. Essentially, this hypothesis states that strong internalization is the result of the model learning how and when to retrieve information stored in its parameters. In our experiments, the model could selectively retrieve information, definitions from $\mathcal{X}_2$, at test time, despite never needing to retrieve those definitions in a similar way during training.

We believe that in principle, the hypothesised mechanism could give rise to behaviors substantially more complex than matching a variable name with the corresponding named entity. This explanation could be studied using the tools of mechanistic interpretability to try to understand if and where definitions are stored, and how they are retrieved. For instance, one might discover circuits (Olah et al., 2020) that inhibit the retrieval of Define definitions, or perhaps perform interventions on the model's activations such that Define definitions are treated by the model like Define ones, or vise versa. Such studies can help precisely understand what is going on inside the model when it better internalizes some specific kinds of data, and generally shed light on how neural nets model the world.

# 5 Related work

**Internal knowledge and world modeling in LLMs.** Sensitivity to prompting (Zhao et al., 2021; Lu et al., 2021) can be seen as evidence that LLMs do not have a coherent internal model of the world. On the other hand, Burns et al. (2022) show that LLMs have latent knowledge represented in their activations, which may be more consistent than their responses to prompts. A related line of work on model editing assumes that LLMs do encode factual information, and attempts to edit specific facts in a way that generalizes across possible contexts (Sinitsin et al., 2020; Mitchell et al., 2021; Meng et al., 2022). Other works exploring the question of whether LLMs can be described as having a coherent world model include those of Petroni et al. (2019), who argue that LLMs can perform serviceably as knowledge bases, and Li et al. (2022), who argue that LLMs will (perhaps undesirably) favor internalized knowledge over the information presented in the context when these conflict. Ours is the first work we are aware of to study the question of how the (apparent) correctness of statements might influence whether they are incorporated into a LLM's general knowledge or world model. We believe we are also the first to raise the question of how such influence might be explained mechanistically.

Andreas (2022) and Janus (2022) suggest that it might not make sense to think of language models as having a single coherent world model since LLMs can simulate a variety of agents, e.g. people, with internally coherent yet mutually contradicting worldviews. In this paradigm, out-of-context meta-learning might help explain how LLMs learn to simulate agents with internally coherent world models, and clarify how LLMs internalize knowledge useful for simulating multiple different agents.

**In-context (meta-)learning.** Brown et al. (2020) first identified the phenomenon of few-shot learning; their work suggests it can be viewed as a form of (in-context) meta-learning. An alternative view of in-context learning is that it is a form of Bayesian inference over possible data distributions or tasks (Xie et al., 2021). Chan et al. (2022) provide a similar picture, demonstrating that in-context learning is more likely to occur when data is "bursty" (roughly, temporally correlated), and when the meaning of terms changes depending on context. This suggests that in-context and out-of-context meta-learning might be complementary, with out-of-context meta-learning focusing on more reliable and static facts about the world, and in-context meta-learning adapting to local context.

**Gradient alignment.** A large number of existing works study or encourage gradient alignment as measured by inner products, cosine similarity, or (negative) $L_2$ distance. This includes works on meta-learning (Nichol et al., 2018; Li et al., 2018), multi-task learning (Lee et al., 2021), optimization (Zhang et al., 2019), generalization (Roberts, 2021), domain generalization (Parascandolo et al., 2020; Shi et al., 2021; Li et al., 2018), implicit regularization (Smith et al., 2021), and understanding deep learning (Fort et al., 2019). However, we are not aware of any systematic survey of gradient alignment, and these works have remained somewhat siloed. Most relevant to our work are those works that focus on meta-learning and implicit regularization of SGD. In particular, Nichol et al.

(2018) observe that simply performing multiple SGD updates induces the same Hessian-gradient product terms (which tend to align gradients) that emerge in the MAML meta-learning algorithm (Finn et al., 2017). Meanwhile, Smith et al. (2021) use backward error analysis to show that SGD implicitly penalizes the variance of gradients across mini-batches (or, equivalently, rewards gradient alignment), with the strength of the penalty being inversely proportional to mini-batch size. While Dandi et al. (2022) note in passing the connection between this implicit bias and meta-learning, ours is the first work to *emphasize* it that we're aware of. We go beyond previous works by demonstrating qualitative differences in learning behavior (specifically, weak and strong internalization) caused by using stochastic (vs. full-batch gradient) gradient methods.

# 6    Potential Implications for Safety of Advanced AI Systems

Understanding and forecasting AI systems' capabilities is crucial for ensuring their medium- and long-term safety. Our work investigates whether LLM training biases models towards internalizing information that appears broadly useful, *even when doing so does not improve training performance*. Such learning behavior might represent a surprising capability which might change designer's estimation of system's potential to do harm. In particular, we believe internalization is a plausible mechanism by which LLMs might come to believe true facts about the world. This might lead them to acquire situational awareness (Ngo, 2022) and obey normative principles of reasoning.

Elaborating on this second concern: One particularly concerning type of normative principle that has been postulated is functional decision theory, which encourages intelligent agents to cooperate with other similar agents (Yudkowsky and Soares, 2017). Cohen et al. (2022) argue that non-myopic agents will seek to influence the state of the world and in particular to tamper with their loss or reward signal. On the other hand, Krueger et al. (2020) argue that while reinforcement learning (RL) agents indeed tend to pursue incentives to influence the state of the world, such incentives may be effectively hidden from systems trained with supervised learning or "myopic" RL (trained to optimize immediate reward by setting the discount rate, $\gamma = 0$). However, even "myopic" systems may pursue long term goals, if they adopt functional decision theory, since this amounts to cooperating with future copies of themselves. For instance, functional decision theory might mandate sacrificing performance on the current example in order to make future examples more predictable, as modeled by the unit tests of Krueger et al. (2020). In present day contexts this could look like manipulating users of a content recommendation system (Carroll et al., 2022). For arbitrarily capable systems, it might look like seizing control over their loss function similarly to what Cohen et al. (2022) describe with RL agents. We are interested in better understanding out-of-context meta-learning so we can either definitively rule out such scenarios (at least those where internalization is part of the mechanism), or take measures to prevent such scenarios.

# 7    Discussion

**Limitations.**    Our work has a number of limitations. Chief among them is the lack of a conclusive explanation for weak and strong internalization. While we discuss two possible mechanisms that could explain internalization, and provide some evidence towards implicit regularization of mini-batch gradient descent playing a role, our understanding of internalization remains incomplete. Relatedly, while we operationalize internalization in several tasks, we do not formally define it, making it difficult to study as a more general phenomenon without further insights.

Furthermore, our LLM experiments were conducted in a multi-epoch training setting, which differs from how these models are typically trained in practice. Nonetheless, our image experiments in Section 3.2 are conducted in a single-epoch setting, and clearly demonstrate the presence of strong internalization. Hence, the phenomenon doesn't appear isolated to the multi-epoch setting.

**Conclusion.**    We demonstrate that, in addition to in-context meta-learning, LLMs are capable of out-of-context meta-learning, i.e. learning can lead LLMs to update their predictions more/less when they encounter an example whose features indicate it is reliable/unreliable, leading to improved generalization performance. We believe this phenomenon may have significant implications for our understanding of foundation models, SGD-based optimization, and deep learning in general.

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
