# A  QA dataset generation

This section describes the creation of datasets used to elicit out-of-context meta learning in LLMs. Code to generate this data can be found at https://anonymous.4open.science/r/internalization-8B46.

In text-based experiments, our data is not IID. The data generating process can be seen in the graphical model in Figure 6. However, the MNIST experiment data is IID; hence this property does not appear necessary for observing the behaviour seen in our experiments.

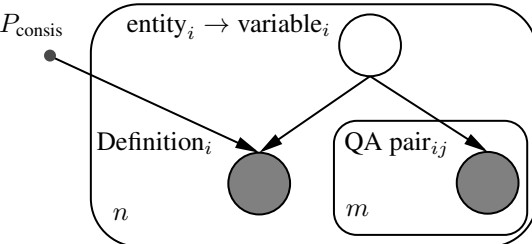

Figure 6: Probabilistic graphical model for dataset creation. $P_{\text{consis}}$ determines the chance that a variable's definition would be consistent with the QA pairs about the same variable.

## A.1  CVDB

We used a Cross-Verified database (CVDB) of notable people 3500BC-2018AD (Laouenan et al., 2022) which includes 2.23m individuals. We removed all names which contain non-alphanumeric characters. Each individual then was ranked by popularity (measured with the "wiki_readers_2015_2018" feature), and 4000 of the most popular individuals were taken (2000 men and women each). We employ 6 types of questions:

1. Gender question: "What was the gender of <name>?". Example answer: "male".

2. Birth date question: "When was <name> born?". Example answer: "19 century".

3. Date of death question: "When did <name> die?" Example answer: "1910s".

4. Question about region: "In which region did <name> live?" Example answer: "Europe".

5. Activity question: "What did <name> do?" Example answer: "actor".

6. Nationality question: "What was the nationality of <name>?" Example answer: "France".

Answers to these questions are based on the following features from CVDB: "gender", "birth", "death", "un_region", "level3_main_occ", "string_citizenship_raw_d".

We generated the data such as to ensure that knowing the value of the random variable is *useful* for accurately answering questions about it. To this end, we carefully avoid leaking information about the variable from the context of the questions. For example, if one of the questions is "When did $xyz$ announce iPhone 4s?", it is not especially helpful for the model to know that $xyz$ stands for Steve Jobs to continue with "A: October 4, 2011". Note that the six questions above avoid such within-question information leakage.

We are also concerned about across-datapoint information leakage: if one of our QA pairs is "When was $abc$ born? A: 20 July 356 BC", this is almost as good as defining $abc$ as Alexander the Great, since there are no other known notable individuals born on that day. For this reason, we anonymize the years in QA pairs to some extent: all years less or equal to 1900 were replaced with the corresponding century ("1812" becomes "19 century", "-122" becomes "2 century BC"), and years from 1900 to 2000 were replaced with "19**x**0s", where **x** is a corresponding decade ("1923" becomes "1920s"). Years greater or equal to 2000 were left unchanged.

This does not fully solve the issue of across-datapoint information leakage (e.g. knowing that someone was born in the 18th century allows one to say that they also died in the 18th or the 19th century), but suffices to make definitions useful enough for our experiments.

## A.2 T-REx

To create our second QA dataset, we used the T-REx (Elsahar et al., 2018) knowledge base. First, we extracted all possible triplets of (subject, predicate, object). Then, we selected the triplets where the predicate is related to creative works, described in Table 1. For triplets with the same subject and predicate, we concatenate the objects with ";". The resulting triplets are converted into QA pairs in accordance with Table 1. Finally, we select QA pairs s.t. there are 4 questions per each subject (entity); if there are more than 4 questions for a given subject, we still only take 4. This is the case for a bit over 6900 entities, which we round down to 6900.

A note on QA pair creation. Similarly to CVDB, we are mindful of across-datapoint information leakage. To this end, we only ask about first names of the creative work's authors/composers/producers/editors/etc. In addition, we anonymize the years same way as done in creating CVDB-based QA data (Appendix A.1).

| Predicate | Question |
|---|---|
| P180 | What does [X] depict? |
| P195 | Which collection is [X] part of? |
| P135 | Which movement is [X] associated with? |
| P123 | Who is the publisher of [X]? |
| P750 | What is the distributor of [X]? |
| P275 | What is the license of [X]? |
| P127 | Who owns [X]? |
| P178 | Who developed [X]? |
| P407 | In which language was [X] published? |
| P364 | In which language was [X] published? |
| P577 | When was [X] published or released? |
| P179 | Which series is [X] part of? |
| P50 | First name of the author of [X]? |
| P57 | First name of the director of [X]? |
| P58 | First name of the screenwriter of [X]? |
| P344 | First name of the cinematographer of [X]? |
| P161 | First name of a cast member of [X]? |
| P162 | First name of the producer of [X]? |
| P1040 | First name of the editor of [X]? |
| P98 | First name of the editor of [X]? |
| P88 | First name of the commissioner of [X]? |
| P86 | First name of the composer for [X]? |
| P136 | What is the genre of [X]? |
| P921 | What is the main subject of [X]? |
| P840 | Where is [X] set? |
| P915 | Where was [X] filmed? |

Table 1: Given a triplet (subject, predicate, object), the question-answer pair is composed by replacing [X] with the subject in the question, and using the object as the answer.

## A.3 Data splits

We split the data into subsets as follows. 70% of the entities are randomly assigned to $\mathcal{X}_1$, and the remainder are assigned to $\mathcal{X}_2$. Then, these entity groups are randomly split into the various subsets of $\mathcal{X}_1$ and $\mathcal{X}_2$ in accordance with Table 2. An entity being assigned to a given data subset means that this subset would include definitions and/or QA pairs corresponding to this entity, and no other subset would include these.

Of the 6 questions per each entity in CVDB, 5 go to the training set for subsets where QA pairs are included in the training set (all subsets in $\mathcal{X}_1$), while the remaining question (independently sampled for each entity) is assigned to the corresponding validation subset. All six QA pairs of each entity go into the test set for $\mathcal{X}_2$. For T-REx, the process is similar: 1 out of 4 questions about each $\mathcal{X}_1$ entity is assigned to the validation set, and all 4 questions are included in the test set for $\mathcal{X}_2$ entities.

|  | Subset | Percent entities |
|---|---|---|
| $\mathcal{X}_1$ | $\dot{\mathrm{D}}_1^{\mathrm{cons}}\mathtt{QA}_1$ | 25 |
|  | $\overline{\mathrm{D}}_2^{\mathrm{incons}}\mathtt{QA}_2$ | 25 |
|  | $\mathtt{QA}_3$ | 10 |
|  | $\hat{\mathtt{QA}}_4$ | 10 |
| $\mathcal{X}_2$ | $\dot{\mathrm{D}}_5^{\mathrm{cons}}$ | 10 |
|  | $\overline{\mathrm{D}}_6^{\mathrm{cons}}$ | 10 |
|  | $\mathtt{QA}_7$ | 10 |

Table 2: Percentage of all entities assigned to each data subset. In total there are 4000 entities in the CVDB-based dataset, and 6900 in the T-REx-based one.

## B  Hyperparameters used when finetuning LLMs on QA data

We use the HuggingFace Transformers (Wolf et al., 2020) library to finetune the LLMs on $\mathcal{X}_1$ for 20 epochs, and on $\mathcal{X}_2$ for 10 epochs. Finetuning on $\mathcal{X}_1 \cup \mathcal{X}_2$ is done for 20 epochs. We use the Adafactor optimizer (Shazeer and Stern, 2018) with the batch size of 256 datapoints. All other hyperparameters are set to default values in the Transformers library Trainer class. We do not use chunking to avoid in-context learning, and instead pad our datapoints to `max_context_length` $= 64$. We use the `deduped` versions of the Pythia models (Biderman et al., 2023).

## C  Additional results from finetuning LLMs on CVDB and T-REx datasets

### C.1  Two-stage results for Pythia-2.8B: entity attribution on CVDB and all T-REx results

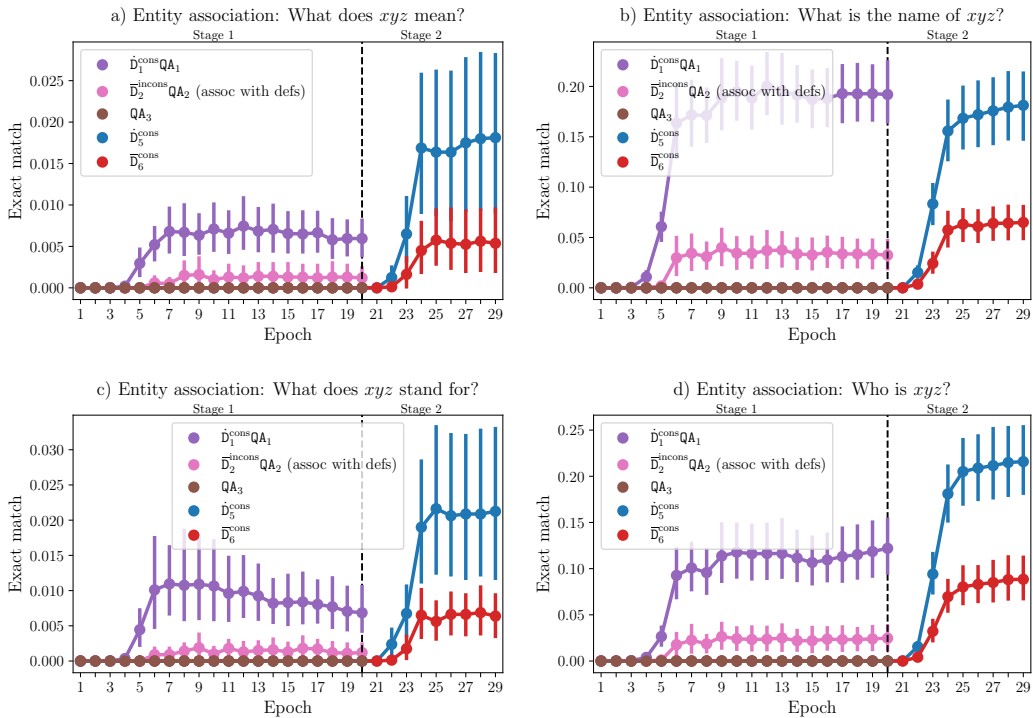

Figure 7: Entity attribution experiments for the Pythia-2.8B-deduped model on the CVDB dataset over 20 seeds. We observe weak and strong internalization for all four question types. Plot b) is the same as Figure 2b in the main paper.

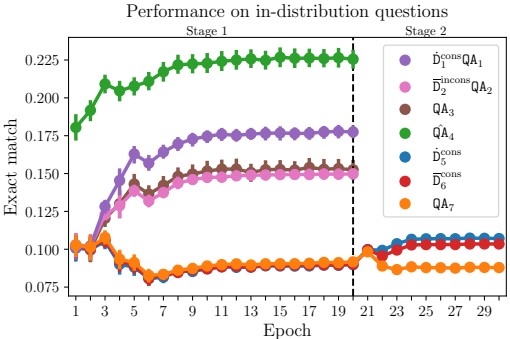

Figure 8: Exact match on the validation subsets for the Pythia-2.8B-deduped model finetuned on the T-REx dataset in two stages over 30 seeds. As with CVDB, we observe weak and strong internalization, albeit strong internalization has a smaller effect than for CVDB (the gap between the blue and the red lines in the second stage is smaller), which we believe is likely because the T-REx dataset is harder.

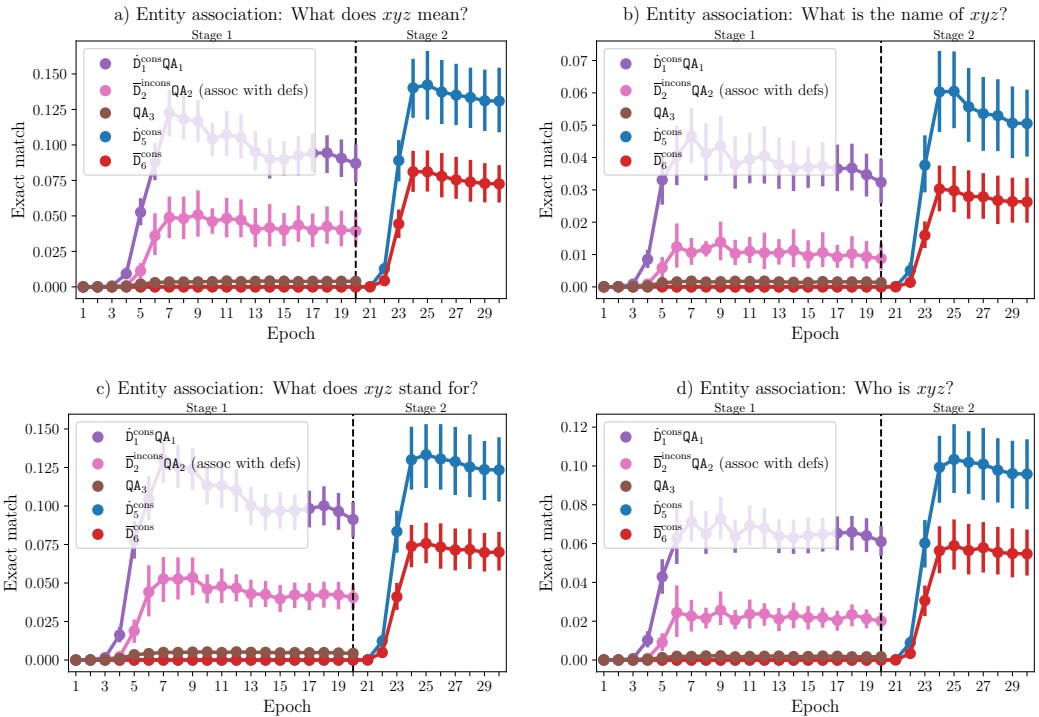

Figure 9: Entity attribution experiments for the Pythia-2.8B-deduped model on the T-REx dataset over 30 seeds. The results appear broadly in line with those observed with the CVDB dataset: we observe weak and strong internalization for all four question types.

 **C.2 Single-stage results for Pythia-2.8B**

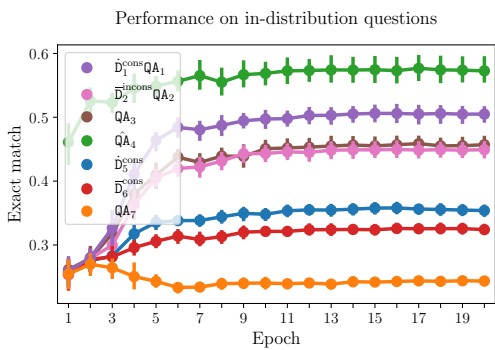

Figure 10: Exact match on the validation subsets for the Pythia-2.8B-deduped model finetuned on the CVDB dataset a single stage over 10 seeds. As with two-stage experiments, we observe weak and strong internalization.

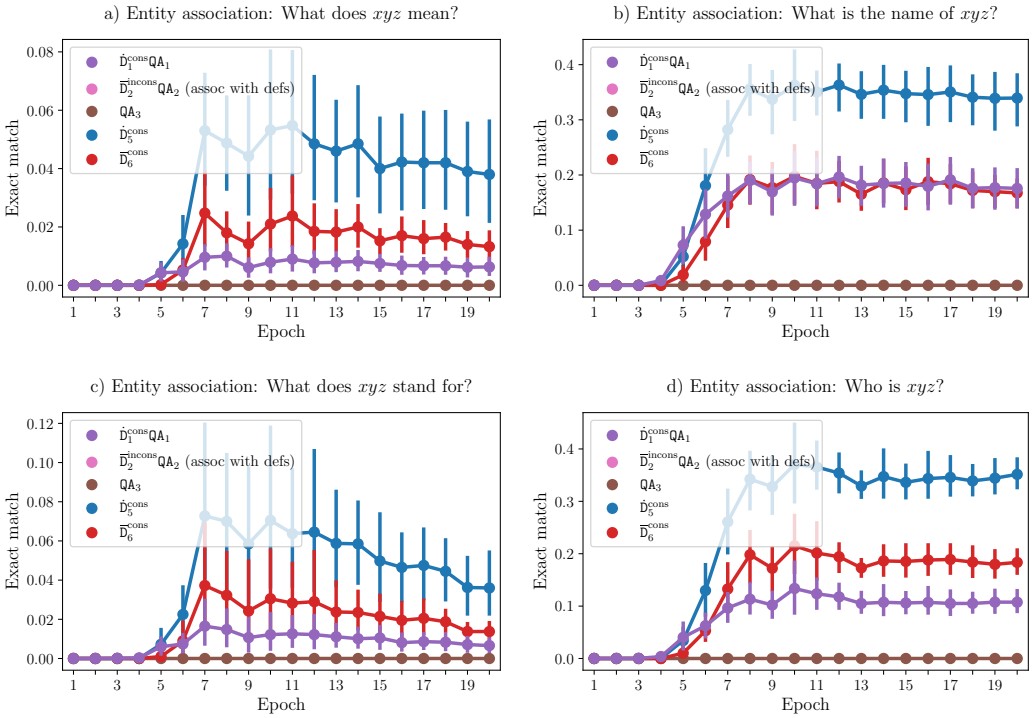

Figure 11: Single-stage entity attribution experiments for the Pythia-2.8B-deduped model on the CVDB dataset over 10 seeds. We observe strong internalization for all four question types. NOTE: this experiment was accidentally launched with $\overline{D}_2^{incons}QA_2$ test set disabled, so we cannot say anything about weak internalization from this.

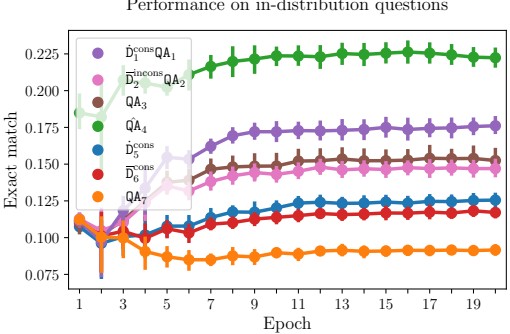

Figure 12: Exact match on the validation subsets for the Pythia-2.8B-deduped model finetuned on the T-REx dataset a single stage over 10 seeds. As with two-stage experiments, we observe weak and strong internalization.

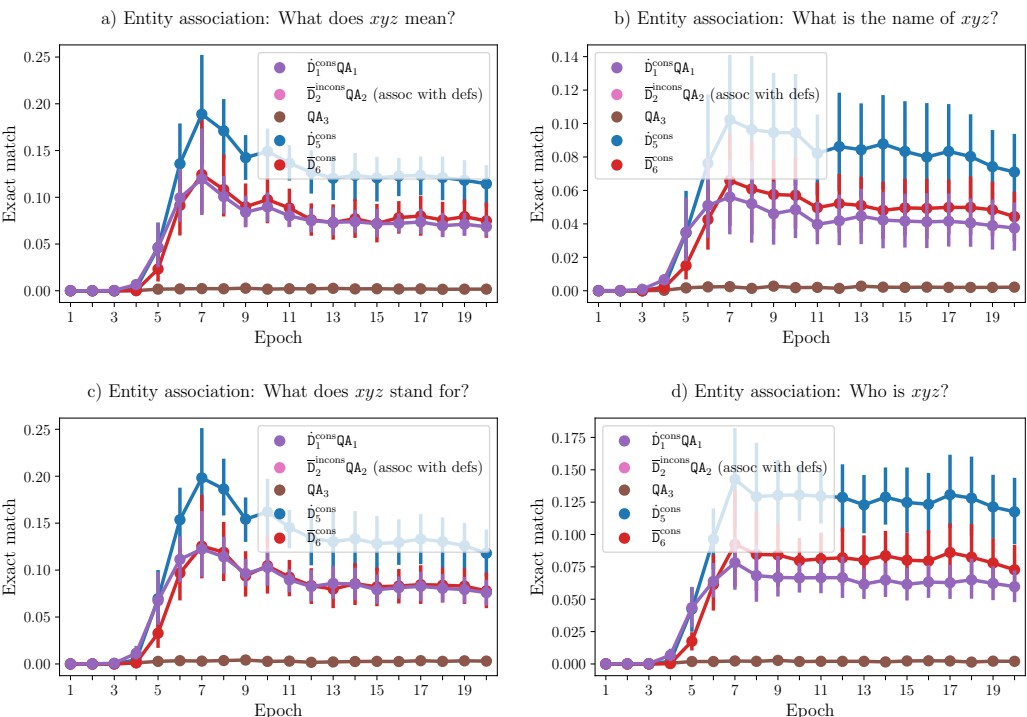

Figure 13: Single-stage entity attribution experiments for the Pythia-2.8B-deduped model on the T-REx dataset over 10 seeds. We observe strong internalization for all four question types. NOTE: this experiment was accidentally launched with $\overline{D}_2^{incons}QA_2$ test set disabled, so we cannot say anything about weak internalization from this.

 ## C.3 Two-stage results for Pythia 410M

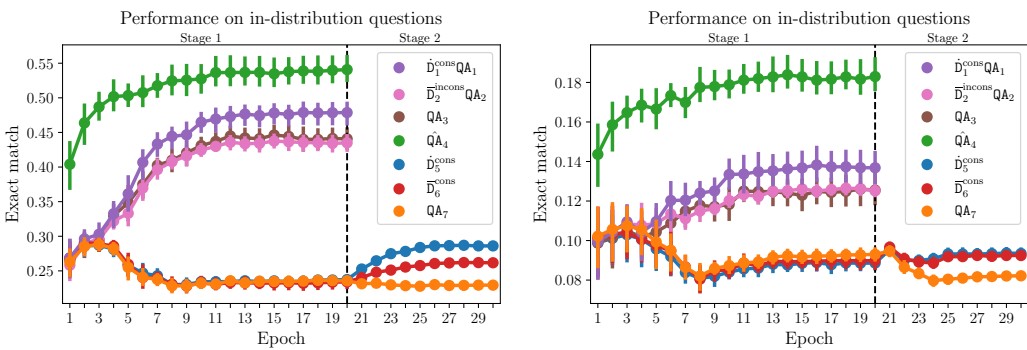

Figure 14: Exact match on the validation subsets for the Pythia-410M-deduped-v0 model finetuned on the CVDB (left) and T-REx (right) datasets in two stages over 10 seeds. We clearly observe weak and strong internalization on CVDB. For T-REx, it appears that the model may be too small to detect strong internalization reliably.

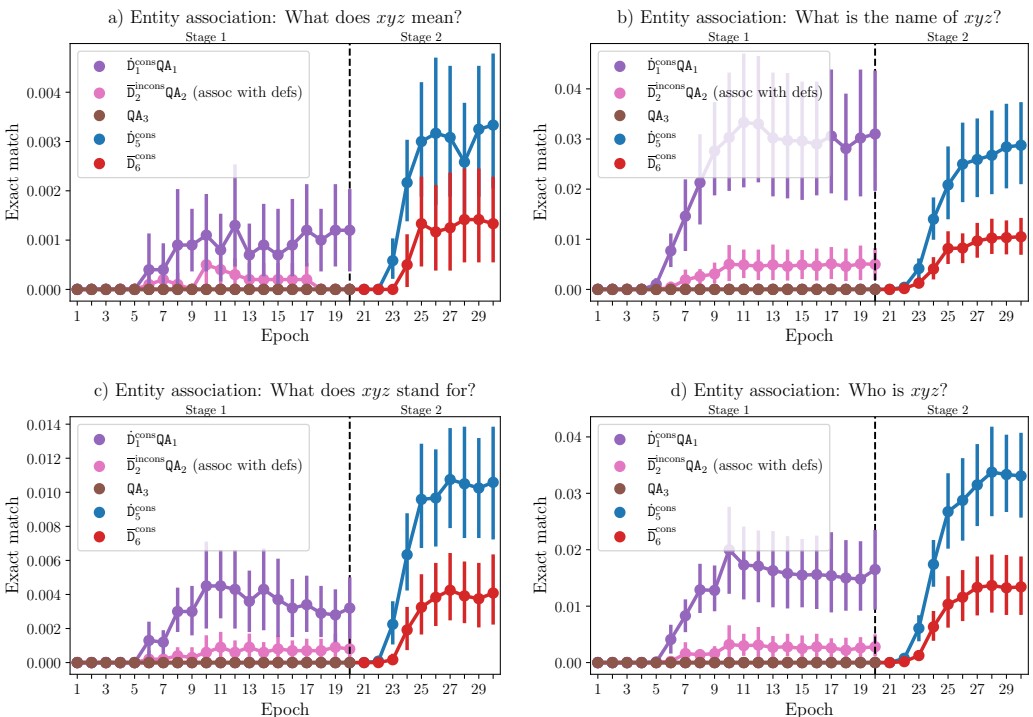

Figure 15: Entity attribution experiments for the Pythia-410M-deduped-v0 model on the CVDB dataset over 10 seeds ($\mathcal{X}_1$ is re-sampled 10 times, and $\mathcal{X}_2$ is re-sampled 3 times per each $\mathcal{X}_1$). The results appear broadly in line with those observed with the larger Pythia model: we observe weak and strong internalization for all four question types. However, the absolute values of EM appear much lower than those of similar experiments with the 2.8B model.

 **C.4    Varying the batch size during single-stage finetuning of Pythia-1B**

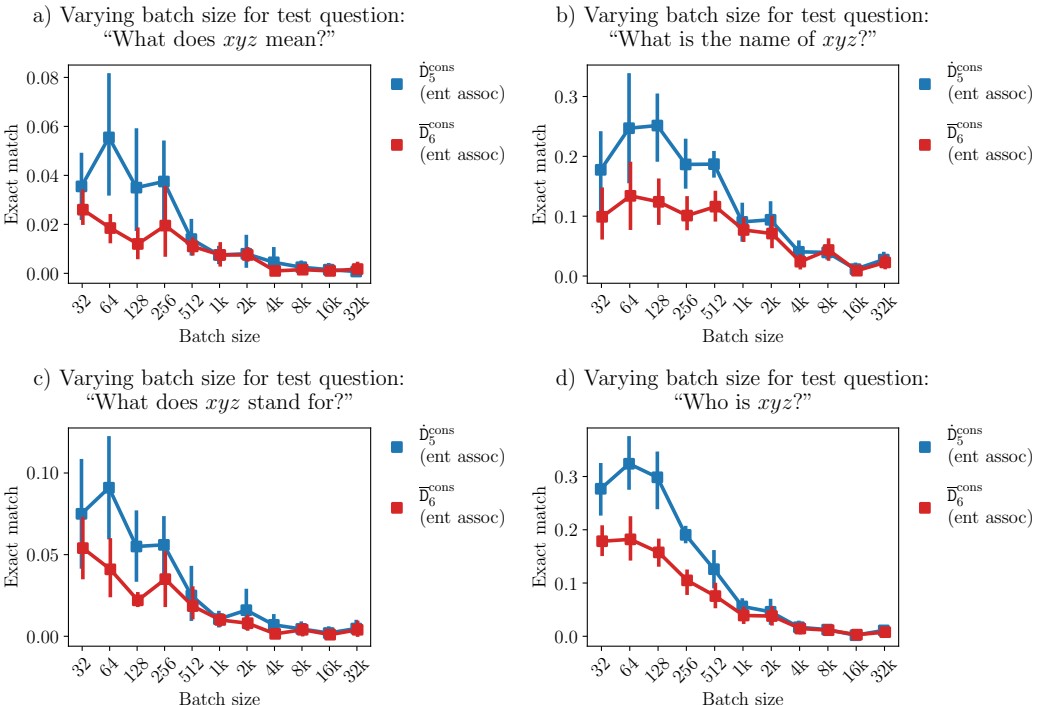

Figure 16: Extent of strong internalization exhibited by the Pythia-1B-deduped model on the CVDB dataset across a range of batch sizes used in single-stage finetuning. Models are trained until convergence over 5 seeds. Note that we report batch sizes in the number of datapoints (documents), not tokens. Larger batch sizes tend to result in less strong internalization; however, this trend might be showing showing signs of reversal at batch size 32. This figure is meant to complement Figure 3c.

 **C.5    Sequence-to-sequence model experiments: setup and results**

To establish the generality of our results, we reproduce weak and strong internalization in a sequence-to-sequence model. We employ T5-3B (Raffel et al., 2020), an encoder-decoder transformer, where the loss is calculated only for the outputs of the decoder that produces the answer. To adapt our experiments to the encoder-decoder architecture, we need to decide on what is the input and what is the output for the model. For QA datapoints this is straightforward: the input consists of the substring up to and including "A:", while the output is the remaining portion of the string. For example, the QA string "Q: what did *xyz* do? A: Queen" gets divided into "Q: what did *xyz* do? A:" and " Queen". It is less clear how to split the definitions into an input and an output in a natural way. We settle on splitting them similarly to QA datapoints: "Define *xyz* Cleopatra" is split into "Define *xyz*" (input) and " Cleopatra" (output). Our results for single-stage and two-stage finetuning are shown in Figures 17 and 18.

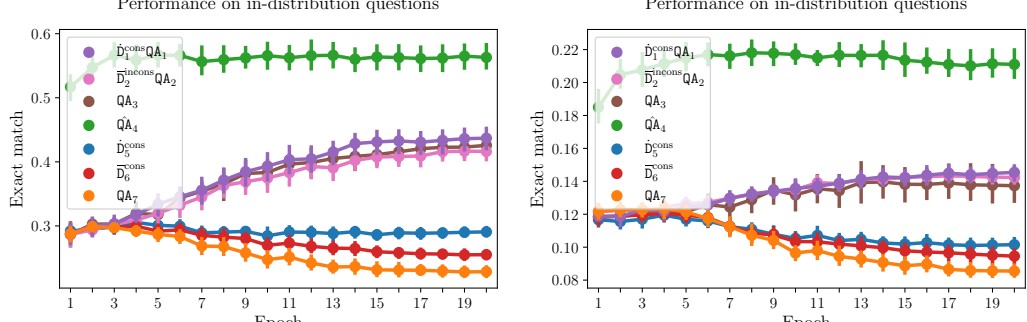

Figure 17: T5-3B finetuned in a single stage on CVDB (left) and T-REx (right) datasets over 10 seeds. The weak internalization effect is seemingly present but barely visible; strong internalization is clearly present.

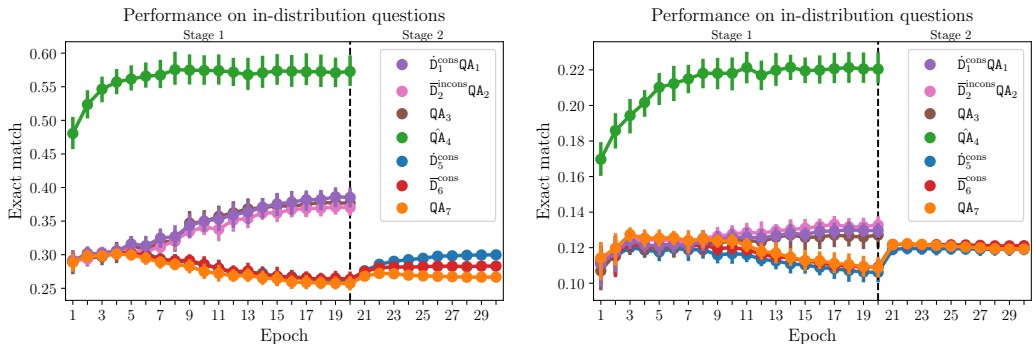

Figure 18: T5-3B finetuned in a two stages on CVDB (left) and T-REx (right) datasets. For CVDB, the weak internalization effect is seemingly present but barely visible; strong internalization is clearly present. For T-REx, looks like neither weak nor strong internalization is present.

## D   Set inclusion experiment

**Data setup.**   Data splits are produced similarly to those in the QA experiment (Sec. A.3), and are summarized in Table 3. We generate test questions such that half of them have the correct answer "Yes" and half "No", hence random guessing would result in 50% accuracy.

| | Subset | Percent variables |
|---|---|---|
| $\mathcal{X}_1$ | $\dot{D}_1^{cons}QA_1$ | 40 |
| | $\overline{D}_2^{incons}QA_2$ | 40 |
| $\mathcal{X}_2$ | $\dot{D}_5^{cons}$ | 10 |
| | $\overline{D}_6^{cons}$ | 10 |

Table 3: Percentage of all variables assigned to each data subset. There are 8000 variable-number pairs in total.

**Hyperparameters.**   We use the Adafactor optimizer (Shazeer and Stern, 2018) with the batch size of 512 datapoints; all the other hyperparameters are Pythia-70m defaults. We train the model from scratch for 100 epochs in the first stage, and for 40 epochs in the second stage.

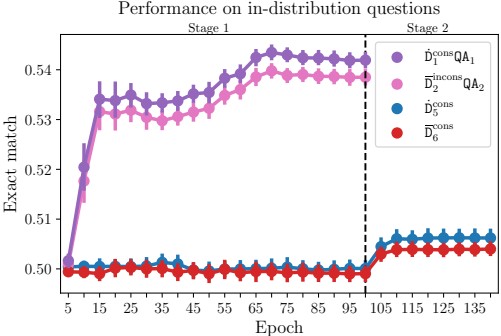

Figure 19: Set inclusion experiment, Pythia-70M model with a custom tokenizer trained from scratch over 50 seeds. We observe both weak and strong internalization. An interesting aspect of this experiment is that if we increase the number of training questions in $\mathcal{X}_1$ per each variable (currently 12), we get much better performance on the validation questions, but the correct definitions stop making a difference.

## E  MNIST experiment

### E.1  MNIST QA Dataset

Here, we give the implementation details for the MNIST dataset, as described in Section 3.2. We used a $3 \times 3$ grid variant of the dataset, yielding $10^9$ possible combinations of digits for the possible values of the variables.

For the training dataset, the digit images to be concatenated into a grid are sampled uniformly at random from all images with the adequate label from the MNIST train split. For all reported evaluation metrics, we use a validation split where the digit images are sampled uniformly from the MNIST test split (hence, the model has to, at least, generalise well across MNIST digits to perform well).

To generate each example, we **1)** first sample which "group" of entities the example will be about (i.e. which of $(\dot{\mathrm{D}}_1^{\mathrm{cons}}\mathrm{QA}_1)$, $(\overline{\mathrm{D}}_2^{\mathrm{incons}}\mathrm{QA}_2)$, $(\mathrm{QA}_3)$, ... in $\mathcal{X}_1 \cup \mathcal{X}_2$, each with equal probability), **2)** whether it will be a definition or a QA example (it's a definition with probability $0.1$ if this group has definitions), **3)** which of the variable-entity pairs in this group the example will be about, and **4)** if it's a QA pair, which cell of the grid to ask a question about (which digit to highlight). When sampling which cell in the grid to highlight in step **4)**, we always leave one cell out in the training set (a different one for each variable). This way, we can also estimate weak internalization, as otherwise the model would achieve perfect accuracy for variables for which it has seen all possible QA pairs in the training set.

At each step of training, we sample a new batch of examples in this way, effectively giving us one-epoch training; in all likelihood, no two examples seen during training will be exactly alike.

The definition pattern, seen in Figure 4(middle) at the top of the definition example, is a uniformly randomly sampled bit pattern for each of the two definition tags, represented as a row of black or white squares (2 pixels each) at the top of the image. The highlight, seen in Figure 4(right), is a 1 pixel wide border around the chosen digit.

### E.2  Hyperparameters for the MNIST QA experiments

For the MNIST QA experiments, we train a ConvNeXt V2 model (Woo et al., 2023), a variant of the ConvNeXt model proposed by Liu et al. (2022). We use the "*Tiny*" variant – a convolutional model with 28.6 million parameters. We train the model with `AdamW` for 120000 training steps with a batch-size of 128, learning rate $3 \times 10^{-4}$, 2000 steps of linear learning rate warm-up, and other optimization hyperparameters matching the original paper.

### E.3 Additional results for MNIST QA Dataset

As mentioned in Section 3.2, we also observe weak internalization in the MNIST QA experiments. The results are shown in Figure 20.

As described in Section E.1, even for the entity groups $\dot{\text{D}}_1^{\text{cons}}\text{QA}_1$ and $\bar{\text{D}}_2^{\text{incons}}\text{QA}_2$ for which QA pairs were present in the training dataset, using definitions is required to get perfect accuracy on the test set, since we never ask questions about one of the grid cells for each variable in the training set. This makes weak internalization apparent in Figure 20.

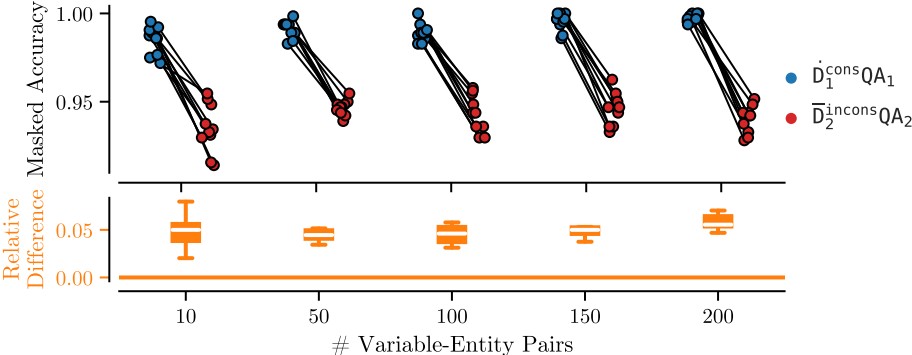

Figure 20: Weak internalization in MNIST QA experiments. Test accuracy on the QA pairs in the validation set for entities in $\dot{\text{D}}_1^{\text{cons}}\text{QA}_1$ and $\bar{\text{D}}_2^{\text{incons}}\text{QA}_2$.

## F  Computational resources used for our experiments

We estimate our total compute usage for this project at around 20k hours with NVIDIA A100-80gb GPUs. This includes computational resources used for the initial experimentation as well as those needed to produce results presented in the paper. Running a single seed of the two-stage CVDB experiment with the Pythia-2.8B model takes about 6 GPU hours. Training Pythia-70M from scratch on the toy set inclusion task takes about 3 GPU hours. Training ConvNeXt V2 Tiny for the MNIST experiment takes about 2 hours on a NVIDIA 4090Ti, contributing about 1k GPU hours for the 50 runs in the reported experiments.