# OpenReview forum: "(Out-of-context) Meta-learning in Language Models"
_NeurIPS.cc/2023/Conference — Submitted to NeurIPS 2023_

### Official Review · Reviewer_yfNH · 2023-06-30

**Soundness:** 2 fair
**Presentation:** 1 poor
**Contribution:** 2 fair
**Rating:** 5
**Confidence:** 3

**Summary:**

This work introduces the phenomenon of out-of-context meta-learning in large language models (LLMs). It demonstrates, through carefully designed experiments, that LLMs have the ability to internalize the semantic content of the text that appears to be from a reliable source.
Specifically, they focus on exploring the existence of weak internalization and strong internalization in the context of LLMs and other vision models.
Potential explanations for the emergence of internalization are explored, including the way models store knowledge in their parameters and the implicit gradient alignment bias of gradient-descent-based methods. Finally, the implications of these findings for future AI systems are discussed, including the potential risks associated with internalization.



**Strengths:**

1. Innovative Methodology: The paper introduces the phenomenon of out-of-context meta-learning in large language models (LLMs) and presents a series of carefully designed synthetic experiments to establish its existence.
The methodology employed in these experiments is unique and provides valuable insights into how LLMs internalize and apply semantic content in different contexts.

2. Comprehensive Experimental Design: The paper describes a series of synthetic experiments that evaluate the phenomenon of out-of-context meta-learning in LLMs from multiple perspectives. The experiments consider different variables and define tags and questions.


3. Implications and Risks: The paper discusses the implications of the findings for future AI systems and highlights potential risks associated with internalization. This analysis adds an important dimension to the paper, emphasizing the importance of understanding and mitigating the challenges posed by out-of-context meta-learning in LLMs.

4. Reproducibility: The paper provides detailed information about the experimental setup, including hyperparameters and performance evaluation metrics.


**Weaknesses:**

1. This work is hard to penetrate. For example, the definition of statements involving two different define tags is not well-defined. Do the statements indicate 'definitions'? Furthermore, the authors' intention behind the phrase 'in every example in which it appears' is unclear.  Additionally, the explanation of weak internalization and strong internalization is confusing. By stating that "LLMs will be likely to respond to questions as if the true statements from the training set are in fact true," do the authors imply that LLMs tend to generate correct answers when variables are defined with consistent define tags?


2. Confusing annotations.

    - In section 2.1, the named entity is represented by a randomly generated 5-character string, whereas Figure 1 shows a 3-character string as the named entity replacement.

    - It would be helpful to use the example presented in Figure 1 for illustration purposes, as it could alleviate comprehension difficulties.

    - The definition of $X_2$ is introduced after its usage, which makes it difficult to understand.

3. The interpretation of experimental results is lacking clarity.

    - The description of 'in the same (inconsistent) definition' in Line 120 is ambiguous.

    - While the authors suggest that usefulness for predicting other datapoints is not the sole reason, they do not elaborate on the meaning of 'usefulness' or identify other contributing factors.

    - What conclusions can be drawn from comparing EM_{test}(QA_4) and EM_{test}(QA_3)? What is the purpose of the authors' explanation in Line 123-129?

    - How should internalization be understood in the context of 'resemblance to useful data'?

    - Is pretraining necessary? In section 3.1, the authors only provide the experiment setups but fail to give a conclusion.

4. The title does not accurately reflect the content, since this work only focuses on LLMs and also explores such a phenomenon in computer vision models.



**Questions:**

1. Which experiments provide evidence for the statement that 'when the information content of two pieces of text is the same, language models ...'?
2. What distinguishes the entity subset of QA_3 and QA_7?
3. How are the experimental results in Figure 2 obtained? Are they generated by fine-tuning on different subsets and testing on the same test set?
4. What's the difference between in-content meta-learning and out-of-context learning?
5. Could you please explain the data set-up for 'assoc with defs'?
6. What are the data splits of D_8QA_8 and D_9QA_9?

**Limitations:**

I do not foresee any potential for negative societal impact from this work.

---

> ### Author Rebuttal · Authors · 2023-08-10
>
> $\let\n\newcommand \n{q}{\mathtt{QA}}\n{\d}[3]{\mathtt{#1 D^{#2cons}_{#3}QA}} \n{\do}{\mathtt{\dot D}}\n{\a}{\alpha}$Thanks for the detailed feedback and for highlighting parts of our presentation you found unclear! We understand that our experimental setup is intricate, and want to make our work as easy to follow as possible. We will address your questions and suggestions below.
> # Addressing the weaknesses
> ## Clarifying interpretation of experimental results
> >The description of 'in the same (inconsistent) definition' in Line 120 is ambiguous.
>
> Here we don't introduce anything new and just refer to the previously established structure of inconsistent definitions. We'll clarify this in the revision.
> >While the authors suggest that usefulness for predicting other datapoints is not the sole reason, they do not elaborate on the meaning of 'usefulness' or identify other contributing factors.
>
> Lines 119-122 suggest that observing words next to one another during training might make the model associate them at test time, which is a factor that could lead the model to internalize inconsistent definitions. Re usefulness, our work treats it as an informal/intuitive concept.
>
> >How should internalization be understood in the context of 'resemblance to useful data'?
>
> Consistent definitions are useful for predicting answers to questions about the defined variable.  In $\mathcal{X}_1$, consistent definitions always use the $\do$ tag, so we say other examples using this tag are "useful-seeming", and our experiments show these $\mathtt{\dot{D}}$ definitions are internalized more.
>
> >What conclusions can be drawn from comparing QA_4 and QA_3?
>
> This comparison (as well as the comparison of $\q_3$ and $\q_7$) tells us that in our dataset, knowing answers to several questions about a variable allows the model to better answer other questions about this variable, but not as well as when the entities are not replaced with the variables.
>
> >Is pretraining necessary?
>
> It's not, see the final paragraph of Section 3.1. We'll revise to make this clearer.
> ## Clarifying annotations
> We fixed $\mathcal X_2$ being introduced after its first usage, and will clarify the discrepancy with 3 VS 5-character variables. Thanks for highlighting these!
> >It would be helpful to use the example presented in Figure 1 for illustration purposes.
>
> We use the Cleopatra example several times throughout the paper. Do you have suggestions for which other examples it'd be helpful to use, and where?
> ## Title
> We chose current title to contrast *out-of*-context meta-learning with *in*-context learning in LLMs. If reviewers agree, we'd consider changing to "Out-of-context meta-learning in neural networks" or similar.
> # Answering questions
> >The definition of statements involving two different define tags is not well-defined. Do the statements indicate 'definitions'?
>
> We refer to statements containing a define tag as definitions. A definition always has one define tag, and if a given variable has a definition, it is always with the same define tag.
> >The authors' intention behind the phrase 'in every example in which it appears' is unclear.
>
> There we mean that a definition establishes the correspondence between a named entity and a variable across all documents in our dataset, not just within that definition.
> >By stating that "LLMs [...]"?
>
> Yes, e.g. when a $\do$ definition establishes a correspondence between `xyz` and Cleopatra, the model is more likely to correctly treat `xyz` as Cleopatra when answering questions about `xyz`.
> >1. Which experiments provide evidence for the statement that 'when the information content of two pieces of text is the same, LMs ...'?
>
> We interpret a definition as a combination of a feature indicating consistency (the define tag) and the information content connecting the variable and the named entity. With this interpretation, our experiments in Sec. 2.4 (strong internalization) provide such evidence.
> >2. What distinguishes entity subsets QA_3 and QA_7?
>
> Finetuning data includes QA pairs about entities from $\q_3$, but not $\q_7$. We'll update the paper with a table clarifying data subsets (see our response PDF).
> >3. Are [Fig. 2 results] generated by fine-tuning on different subsets and testing on the same test set?
>
> It's the opposite; we train on the same data and different curves correspond to different test subsets.
> >4. What's the difference between in-context meta-learning and out-of-context learning?
>
> In-context meta-learning is when a model "learns" to perform a task simply by conditioning on examples in the prompt. Out-of-context meta-learning is when the model learns to internalize datapoints differently due to previous training (two kinds of definitions in our case). This resembles meta-learning methods like MAML [1], but without any explicit meta-learning algorithm. We'll add a new in-context learning baseline to clarify this contrast (see PDF).
> >5. Could you please explain the data set-up for 'assoc with defs'?
>
> The (assoc with defs) evaluation measures how well the model internalized the contents of inconsistent definitions. Normally, if our inconsistent definition states that e.g. `abc` is Socrates, answers to questions about `abc` will be considered correct if they are **not** about Socrates. In the (assoc with defs) case, we expect answers as if `abc` is in fact Socrates.
> >6. What are the data splits of D_8QA_8 and D_9QA_9?
>
> $\d{\dot}{}{1}$ and $\d{\bar}{in}{2}$ subsets each contain datapoints about $N$ entities. To create $\d{\dot}{in}{8}$, we make $(1-\a) N$ of $\d{\dot}{}{1}$ definitions inconsistent, so the data split between entities with consistent and inconsistent $\mathtt{\dot D}$ definitions becomes $\a N$ for $\d{\dot}{}{1}$ and $(1-\a) N$ for $\d{\dot}{in}{8}$. The split between entities with $\mathtt{\bar D}$ definitions is the opposite, $\a N$ inconsistent and $(1-\a) N$ consistent.
>
> [1] Finn, C, et al. "Model-agnostic meta-learning for fast adaptation of deep networks." ICML 2017.

---

> > ### Comment · Reviewer_yfNH · 2023-08-12
> >
> > Thank you for your time and effort in addressing my major concerns.
> >
> > > We use the Cleopatra example several times throughout the paper. Do you have suggestions for which other examples it'd be helpful to use, and where?
> >
> > I mean that would be better to directory use Cleopatra example in Figure 1 to illustrate the definition of define tags, rather than James Bond example.

---

> > > ### Author Response · Authors · 2023-08-12
> > > **We agree and will revise this**
> > >
> > > That's a good suggestion, thanks! We'll revise this. In a similar vein, we will also replace Shakespeare / William Shakespeare with Darwin / Charles Darwin on line 102.

---

### Official Review · Reviewer_1KMo · 2023-07-07

**Soundness:** 3 good
**Presentation:** 3 good
**Contribution:** 3 good
**Rating:** 6
**Confidence:** 4

**Summary:**

The paper shows the existence of a phenomenon that the authors refer to as out-of-contect meta learning in large language models. The authors design experiments that show that this phenomenon causes the internalization of text that is broadly useful, meaning that the LLM is more likely to treat this content as true. The paper shows two forms of internalization, namely weak and strong internalization, the later being a form of meta learning. Two reasons are suggested for this phenomenon, one based on the parameters of the model, and another one relying on the implicit gradient alignment bias of gradient-based optimization methods.

**Strengths:**

* The paper shows an interesting phenomenon
* The proposed explanations are sound and intersting
* The paper is well written and easy to follow

**Weaknesses:**

* There is no conclusive explanation of the reasons why internalization happens
* The phenomenon is hard to formalize and study, which limits the advantage of the insights in the paper

**Questions:**

None

**Limitations:**

The authors describe limitations in the paper

---

> ### Author Rebuttal · Authors · 2023-08-09
>
> Thank you for you review. We are glad you found the phenomenon and our explanations for it interesting and sound, and our paper well-written and easy to follow.
>
> We also believe we're in full agreement on the limitations of this work, given that they overlap with our limitations section. We think these would be the next open questions to pursue as follow-ups to this work. Since you recommended an accept for our paper, it appears we likely agree that the existence of internalization is, however, worth highlighting to the deep learning community in order to facilitate investigations into it on a broader scale.
>
> If you have any other suggestions for improvements, we'd be keen to take them on board!

---

### Official Review · Reviewer_pjae · 2023-07-08

**Soundness:** 2 fair
**Presentation:** 4 excellent
**Contribution:** 2 fair
**Rating:** 6
**Confidence:** 2

**Summary:**

The paper argues for the existence of ‘out-of-context meta-learning’ as a characteristic of LLMs. The authors support this claim with cleverly designed experiments on QA using a 2.3B parameter pretrained Pythia model. They argue that the presented experiments demonstrate out-of-context meta-learning. They perform additional LLM experiments and a pair of simple toy experiments a synthetic language task and modified MNIST task. The authors discuss hypotheses which may explain the mechanism of their proposed phenomenon, and discuss its implications for the research community at large.

*after rebuttal, updated score from 4 (borderline reject) to 6 (weak accept)*

**Strengths:**

The paper is well structured, flows nicely, and is clearly written. The experiment setup is motivated to test a specific hypothesis, and is highly creative, and the experiments are thoroughly analyzed. Many additional experiments were run. The results have tight error bars and seem likely to be correct and reproducible. The analysis of the implications of the central claim of the paper touches broadly on the capabilities of LLMs and is highly relevant to the general research community, especially as pertains to safety.

**Weaknesses:**

Issue with the central claim
---------
The central claim of the paper is that the model is sensitive to the appearance of authoritativeness / usefulness of specific examples, and incorporates that assessment into some decision as to how thoroughly to 'internalize' those example.

*    “is, or appears to be, *broadly useful* (such as true statements, or text from *authoritative sources*)” line 7
*    [the model] “pick[s] up on features that indicate whether a given data point is *likely to help reduce the loss on other data points*, and “internalize” data more or less based on these features” lines 16-17
*    “Thus *usefulness for predicting other datapoints* is not the only reason why a definition might be internalized." 121-122
*    “So after finetuning on X1, the neural net ends up at a point in the parameter space where gradient updates on consistent-seeming definitions result in more internalization than updates on inconsistent-seeming definitions. We consider this out-of-context meta-learning; it is as if the neural network “expects” the definitions with [blue,dotted]Define to be more useful for reducing the training loss in the future, and thus internalizes them more.” lines 137-143
*    “Our work investigates whether LLM training biases models towards internalizing information that *appears broadly useful*, even when doing so does not improve training performance” line 342
*    “learning can lead LLMs to update their predictions more/less when they encounter an example *whose features indicate it is reliable/unreliable*“ line 377

This is an extraordinary claim, as it supposes capacities of the model that are not immediately obvious in model behavior or in potential underlying mechanism (sec 4). The authors seek to demonstrate this behavior with the QA experiments in section 2. The experiments presented are thorough and interesting, but it is not clear to me that the results they show need to be interpreted as grandly as the authors do. It seems plausible that a simpler explanation may sufficiently explain the observed data without relying on imbuing the model with surprising new capacities.

Potential alternative explanation:
In Section 2, in Figure 2, the authors present the main evidence for their claims. For this argument, Let us suppose the 5-char sequence for the “inconsistent” tag ‘redDEFINEbar’ is “*YUIOP*”, and that the 5-char sequence for the ‘consistent’ tag ‘blueDEFINEdotted’ is “*GHJKL*”. This helps to ground these strings in how the model sees them as opposed to how they may be interpreted.

Incorporating a *GHJKL* sequence, as in QA1, gives the model the opportunity to recover from the loss of information in the entity-string masking (shown in the gap between QA4 baseline and QA3), but only through the medium of updating the parameters of the model themselves (as opposed to via the activations as in in-context learning). QA2 obfuscates further from QA3 by incorporating a *YUIOP* sequence which connects each entity-string to a random incorrect entity. In essence, the *GHJKL* sequences tell the model that the entity-string and entity in the sequence are identical. However, it is not clear to me that the *YUIOP* sequences should be interpreted as “inconsistent seeming [definitions]” (line 138). There is nothing that forces the model to view *YUIOP* as an indicator of identity and then to figure out that its an unreliable identity indicator (which would involve the kind of self-reflection capacities supposed in the claim of the authors). Is it not more parsimonious to say the *YUIOP* sequences are consistent markers of non-identity - simply non-sequitur statements which are true but generally useless? If it is always true that the entity-strings and entities in *YUIOP* sequences are inconsistent with the QA examples holding those entity-strings (‘perfectly correlated’ line 87), a reasonable pattern that the model may learn is “'*YUIOP* X Y' means that X!=Y”. This bizarre anti-definition is almost useless as Y could be anything other than X, and potentially confusing, which can account for the drop from QA3 (brown) to QA2 (pink) following similar reasoning as in lines 120-121. So far this is basically the same, but from this perspective, the “surprise” result of Figure 2, that D5 outperforms D6, is no longer surprising. It is not necessary to rely upon the suggestion that the model “pick[s] up on features that indicate whether a given data point is likely to help reduce the loss on other data points, and “internalize[s]” data more or less based on these features” lines 16-17. Is it not simpler to suggest that the model has learned correctly that *GHJKL* indicates identity and *YUIOP* indicates non-identity? In this case, the fact that non-identity is ‘internalized’ to a lesser degree is no surprise at all: non-identity is only loosely incorporated (or incorporate-able!) because it is a non-sequitur. The model can fail to 'internalize' this non-sequitur information on account of it's general irrelevance, without relying on an surprising capacity to learn conditional on an example's "*usefulness for predicting other datapoints*" (line 122). A similar explanation can be given if *YUIOP* is not understood to be non-identity at all, but just random noise with no consistent interpretation. The fact that the model 'internalizes' the *GHJKL* information more than that of the *YUIOP* can rely solely upon the fact that an interpretation of *GHJKL* is readily apparent and there is no obvious interpretation of *YUIOP*. This line of reasoning begs the question as to the loss curves of the specific *GHJKL* and *YUIOP* examples in QA1/QA2 over the course of the training. You might expect to see higher loss for the *YUIOP* examples. Note that this argument extends to the non-QA experiments presented as well.

What is strange in this perspective is not that D5 outperforms D6 but that D6 outperforms QA7! But this surprising result does not carry the significant implications of the previously surprising result highlighted by the authors. Regardless of how you explain the superior performance of D6 over QA7, it bears explaining why the above reasoning (which explains away the 'surprise' of the gap between D5 and D6 and which does not stipulate any particularly surprising characteristics on the behalf of the model) is confused. It seems a plausible enough explanation that without a convincing rebuttal the central claim of the paper, which makes an extraordinary claim of model behavior, seems shaky.

There is no reason to presuppose that a *YUIOP* example would ever be interpreted as a definition by the model, despite it being labeled as such in the analysis of the paper. Without this presupposition, the claim that *YUIOP* represents an "inconsistent-seeming definition" to the model is unfounded, as is the subsequent claim that that the model “pick[s] up on features that indicate whether a given data point is *likely to help reduce the loss on other data points*, and “internalize[s]” data more or less based on these features”. The gap supposed to demonstrate 'strong internalization' can be explained as nothing more than the difference between the model comprehending a useful control sequence marking identity, *GHJKL*, and a sequence marking random noise *YUIOP*.

(Note: the above arguments may indeed be plausible but subtly misguided and ultimately wrong, but the authors must address them convincingly in order to strengthen the paper.)

Other
-----
Lines 79-82 discuss ‘information leakage’ where replaced entities may be inferrable based on information present in the QA pairs and background information present in the pretrained model, and states that steps have been taken to reduce this possibility. Presumably the performance of QA3 in Fig2 would vary significantly with this information leakage, where highly ‘leaked’ entities would still have good performance? (Ie. training on “Q: xyz was the first president of which country. A: the USA” should yield better performance on xyz related test Qs than a more obfuscated relation). Is this interpretation correct? If so it seems that the function of this information leakage and the specific means of alleviating it are actually very important to the interpretation of the results, and should perhaps be given more attention than being left to the appendices / alluded to in lines 124-127.

Internalization is not formally defined in any way, yet it is a central aspect of the paper. It is 'measured' only via aggregated loss on each dataset. More time should be spent investigating and developing the idea of internalization (how does the 'internalization' of a specific example relate the loss on that example?).

Nits
-----

*    The title of the paper comes from a contrast to ‘in-context’ learning, which is referenced many times in the paper, but the meaning of the term is not made explicit until the Related work (line 314). It would improve clarity to define what is meant by in-context learning and to describe how the proposed ‘out-of-context’ learning differs when introducing the concept of out-of-context learning (line 42).

*    Figure 1 bottom right has a typo: “Q: What did qwe born? A:” is presumably a mish-mash of two different questions? And not actually in the test set? It would not be surprising to get a bad answer to this question as stated!
*    It would be helpful to label the data presented in Figure 1 with the dataset names (QA3, QA4..) used in Section 2.3.
*    Figure 1 depicts two stages of finetuning on two separate datasets X1 and X2, and their subsequent eval/analysis, but this is a little obfuscated by the presentation. Consider making it more organizationally clear. Perhaps draw a bubble around the box on the left and the box on the top right to show they are the same stage, and add a Train label to the dataset in the left box to be consistent with the others. Some explicatory text can be moved to the caption to make the figure itself easier to cartoonify.
*    Line 47 his -> this
*    Fig2: consider showing “the entities consistent with the QA pairs; the latter get accuracy 0 everywhere” (Line 153) in the plot as well
*    Line 291 vise-> vice
*    Consider adding the number of (entity, entity-string) pairs present in the datasets of Fig2. It might be helpful to add a table with each of the datasets presented in Fig2, showing their characteristics and size / number of entity - string pairs. This would help clarity / readability.


**Questions:**

Suggestions:
--------
*   Generally, more time should be spent on justifying the central claim. It is extraordinary, and if true, the authors perspective that it "may have significant implications for our understanding of foundation models, SGD-based optimization, and deep learning in general" seems at least plausible. However, the current text spends little time on justifying this argument, and no time addressing alternative explanations, like that presented in the weakness section. As the paper stands, the experiments as presented do not convincingly support (to the exclusion of simpler explanations) the central claim of the paper.
*    It would help to understand what is going on in model training to present and analyze the loss curves of specifically the *GHJKL* and *YUIOP* examples, for datasets QA1/2 and D5/D6 in the second round of finetuning. The loosely defined 'internalization', which is measured through overall dataset performance, seems like it might be inversely correlated to example perplexity. As perplexity can be measured on a example granularity, this should be presented.
*    It would be nice to demonstrate in the Sec2/Fig2 experiments the result of recovering from the string-masking via *in-context* learning. This is an important baseline that is conspicuously missing (even if it just recovers fully the QA4 performance).



**Limitations:**

The mentioned limitations are well selected (formalization of internalization, absence of obvious mechanism). The rebuttal of alternative explanations for the presented results is absent from the paper, a significant additional limitation. The potential social impacts are discussed.

---

> ### Author Rebuttal · Authors · 2023-08-09
>
> Thank you for your very thoughtful, clear, and in-depth response.  We want to ensure that our work is clear to you and to other readers, and are eager to address any issues with the current framing and experiments. We also want to ensure we understand your position correctly.
>
> We believe your simpler explanation is actually consistent with our central claim -- whatever understanding the model has of `YUIOP`, examples with this tag are not useful, whereas `GHJKL` examples are.  Our experiments demonstrate that the model is sensitive to this difference in usefulness and internalizes (or fails to) accordingly.
>
> ## Clarifying our claims
> > Without this presupposition, the claim that `YUIOP` represents an "inconsistent-seeming definition" to the model is unfounded, as is the subsequent claim that that the model “pick[s] up on features that indicate whether a given data point is likely to help reduce the loss on other data points, and “internalize[s]” data more or less based on these features”
>
> The first claim, that "`YUIOP` represents an 'inconsistent-seeming definition'" is informal -- what what we mean by "definition" is just "example that begins with a define tag"; we will rephrase to reduce confusion.  We also reran the Figure 2 experiments with definitions of the following format: "`GHJKL` According to many texts, `bgn` refers to Darwin" and "`YUIOP` According to many texts, `qwe` refers to Curie", and achieved ~identical results.  Does this help address your concern?  If not, can you please clarify why?
>
> The second claim is central to our work, and one we are happy to defend.  Just to be clear about why we believe it is well supported:
> - The features referenced here are `GHJKL`/`YUIOP`.
> - Due to the way we construct the datasets, internalizing `GHJKL` examples (but not `YUIOP` examples) is likely to reduce loss on corresponding QA pairs.
> - Our evaluation on $\mathcal X_2$ shows that the model picks up on this difference between `GHJKL`/`YUIOP` statements and internalizes `GHJKL` statements more.
>
> Note that this claim and its justification do not rely on interpreting any examples as "definitions".
>
> ## On surprising-ness of your explanation
> We believe it is reasonable to interpret our results as «in the first finetuning stage the model learned that `GHJKL` means “is” and `YUIOP` means “isn’t”, or “identity/non-identity”. Then, in the second stage, the model is finetuned on new sentences essentially of the form:
> - “`bgn` is Charles Darwin”
> - “`qwe` isn’t Marie Curie”
>
> and correctly internalizes entity-variable associations marked with “identity” more than those marked with “non-identity”».
>
> We'd argue that even with this lens, it is non-obvious that a gradient update on “`bgn` is Darwin” and “`qwe` isn’t Curie” doesn’t just make the model more likely to produce these specific strings, but makes it better internalize that `bgn` is Darwin – i.e. to changes it's predictions on novel examples about `bgn` as if `bgn` was Darwin. We emphasize that training loss does not explicitly encourage such internalization (since there are no QA pairs about `bgn` in the training set).
>
> Perhaps you disagree that this is surprising, but this begs the question, why should the model internalize statements about identity (i.e. generalize to different types of examples, such as QA pairs), instead of just learning to predict them and similar statements? We think that it does so is an interesting and novel finding, but welcome further discussion or pointers to related work.
>
> We do suggest out-of-context meta-learning could lead models to develop situational awareness in some contexts, but we don't posit any self-reflection capacity in LLMs on the basis of our results, and make this explicit in our revision.  We also emphasize that the hypothesized mechanisms of internalization presented in Section 4 are fairly mundane.
>
> ## Addressing suggestions
> **Justifying the central claim.** We attempted to clarify and justify our central claim above and in the revision. We currently view your explanation as compatible with ours; we are not sure how they might be empirically distinguished, and welcome further discussion on this.
>
> **Loss curves.** See PDF for per-data-subset training & validation loss curves that we'll include in the revision. Aggregate losses on the training QA samples from $\mathtt{\dot{D}^{cons}_1QA_1}$ and $\mathtt{\bar{D}^{incons}_2QA_2}$ are ~equal, so we are not seeing a higher training loss on `YUIOP` examples. The validation loss **is** higher, as reflected in the EM scores of Figure 2a.  This is consistent with perplexities of individual validation examples being inversely correlated with internalization, as you suggested.
>
> **A baseline with in-context definitions.** We ran this experiment (see PDF), and will include results in the revision. The model indeed recovers performance of $\mathtt{\hat{QA}_4}$ with $\mathtt{\dot{D}^{cons}_1QA_1}$.
>
> ## Other
> Thanks for pointing out the typos (fixed) and the many other helpful suggestions!
>
> **Figure 1.** We added "Train" on top of $\mathcal{X}_1$, and made a clearer visual distinction between the two LM finetuning stages.
>
> **Table describing our data.** We will include this. The response PDF includes an initial version, and we'll add a more detailed version in the appendix.
>
> **Information leakage.** We believe your interpretation is correct, but don't have any experiments that explicitly confirm it. That is, we believe the extent of across-question information leakage should affect the performance on all of {$\mathtt{\dot{D}^{cons}_1QA_1}$, $\mathtt{\bar{D}^{incons}_2QA_2}$, $\mathtt{QA_3}$}. The more leakage there is, the better the performance should be, with the upper bound being $\mathtt{\hat{QA}_4}$.
> We'd be surprised if the gap between the performance on $\mathtt{QA_3}$ and $\mathtt{\hat{QA}_4}$ subsets that we think is needed for internalization was reduced substantially without our mitigations. However, we will report results on this in the appendix.

---

> > ### Comment · Reviewer_pjae · 2023-08-14
> >
> > Thanks for the thorough response!
> >
> > Clarifying
> > ---------
> > > The first claim, that "YUIOP represents an 'inconsistent-seeming definition'" is informal -- what what we mean by "definition" is just "example that begins with a define tag"; we will rephrase to reduce confusion. We also reran the Figure 2 experiments with definitions of the following format: "GHJKL According to many texts, bgn refers to Darwin" and "YUIOP According to many texts, qwe refers to Curie", and achieved ~identical results. Does this help address your concern? If not, can you please clarify why?
> >
> > This does make more explicit that the control sequences are definitions, which addresses concerns about the laxity of referring to them as definitions in the text. My main concern that the control sequences can be most easily interpreted as identity/non-identity remains relevant, though with slightly modified semantics.
> >
> > Surprisingness
> > ---------
> > > We'd argue that even with this lens, it is non-obvious that a gradient update on “bgn is Darwin” and “qwe isn’t Curie” doesn’t just make the model more likely to produce these specific strings, but makes it better internalize that bgn is Darwin – i.e. to changes it's predictions on novel examples about bgn as if bgn was Darwin. We emphasize that training loss does not explicitly encourage such internalization (since there are no QA pairs about bgn in the training set).
> > > Perhaps you disagree that this is surprising, but this begs the question, why should the model internalize statements about identity (i.e. generalize to different types of examples, such as QA pairs), instead of just learning to predict them and similar statements? We think that it does so is an interesting and novel finding, but welcome further discussion or pointers to related work.
> >
> > I do still disagree that this is surprising, so I'll take you up on the question "why should the model internalize statements about identity instead of just learning to predict them and similar statements? Your question is closely related to the following: "why should the model internalize anything at all, instead of just parroting its input sequences directly, and falling into randomness when confronted with novel sequences?" I agree that these are difficult questions to answer well, but fortunately we don't need to do so to explain away the surprising-ness of the model internalizing the true statement 'bgn is Darwin' more than the non-sequitur 'qwe isn't Curie'.
> >
> > It is well known that language models trained only on next-token prediction can develop totally fluent linguistic capacity and draw on a worldview (set of facts/beliefs about the world) that is consistent with but not explicitly encoded in their training data. 20 years ago this last statement would probably have been a surprising and contentious claim (as the robust evidence for it did not yet exist), but it is currently a universally appreciated truth, driven home particularly strongly by the incredible performance of recent LLMs. Given the demonstrated capacities of LMs (especially recent LLMs), the fact that LMs can synthesize related information and make inferences to connect related data points would seem not surprising but an absolutely necessary central component of LMs being able to do any of the interesting things they are currently well-appreciated to be capable of doing.
> > I don't see how the 'internalization' argued for in this work is any different from the already-well-appreciated (if poorly understood) synthesizing capacities of LMs. As this mysterious synthesizing capacity is at the core of how LMs work, I agree it is highly interesting, but not that the presented work has introduced anything novel in pointing out what is referred to as 'internalization'. The explicit exploration of that capacity in particular is a novel contribution, but not, as is claimed, the existence of that capacity at all.
> >
> > For me to believe this is a surprising result would require an understanding of how this 'internalization' is categorically different from the general synthesizing capabilities of next-token-predicting LMs. As it stands, the evidence provided in this work (namely that the true identity statements are internalized greater than non-sequitur statements) I would take to be precisely what one would expect from such a setup.
> >
> > Everything else
> > --------
> > Thanks for addressing the other comments/nits etc. I do think that the experiments on information leakage are central to the interpretation of the results and look forward to an exploration of that in the appendix.
> >
> > Overall
> > -------
> > I still have significant reservations about how the presented results are being interpreted, and welcome a further reply from the authors if time allows. On balance, I think this is a really interesting work, and the community would benefit from its publication and ensuing discussion. I will update my score from 4 to 6.

---

> > > ### Author Response · Authors · 2023-08-18
> > >
> > > Thanks for another very thoughtful response! We agree that recent LLM capabilities are hard to see in any light other than that the models don’t just imitate shallow word co-occurrence statistics, but to some extent understand and internalize the semantic contents of the training data (although there is some debate [1]). We know of only a small number of attempts to study this, e.g. [2] and [3], and believe our work is the first to investigate how *training* on a new datapoint changes the model’s downstream predictions based on the semantic content of this datapoint. Our belief is that the existence of strong internalization will be surprising to a substantial portion of the ML community, and our experiments could help better understand the synthesizing capabilities of LMs.
> > >
> > > We’d be happy to tweak our framing from “we establish the existence of out-of-context meta-learning” to “we explore this phenomenon” if we could cite a work clearly establishing something similar; do you have anything in mind for this? Alternatively, if you perhaps think such a change is warranted since “LLMs internalizing semantic contents of the training data” is ~common knowledge (we're not sure it is) – we could also consider this alteration. We’d be happy to consider other framing changes as well, if you or other reviewers have suggestions.
> > >
> > > [1] Mitchell, M., & Krakauer, D. C. (2023). The debate over understanding in AI’s large language models. Proceedings of the National Academy of Sciences, 120(13).
> > >
> > > [2] Li, B. Z., Nye, M., & Andreas, J. (2021). Implicit representations of meaning in neural language models. Proceedings of the 59th Annual Meeting of the Association for Computational Linguistics.
> > >
> > > [3] Li, K., Hopkins, A. K., Bau, D., Viégas, F., Pfister, H., & Wattenberg, M. (2023). Emergent world representations: Exploring a sequence model trained on a synthetic task. International Conference on Learning Representations.

---

### Official Review · Reviewer_gp6r · 2023-07-13

**Soundness:** 3 good
**Presentation:** 4 excellent
**Contribution:** 4 excellent
**Rating:** 7
**Confidence:** 3

**Summary:**

The authors assert that "out of context metalearning makes LLMs better at internalizing useful information for understanding." They intuitively frame understanding as "treating content as true in question answering." They analyze its application to tasks such as mapping novel phrases or words to attributes and then performing question answers.

They introduce "define tags" which perform the mapping of novel info rather than using the word "define" and natural language, which I like a lot as an approach. This allows them to isolate the effect of the metalearning as a mechanism for improving performance rather than being clouded by the existing notions of the meaning of "define" that may be acquired by the LLM during pretraining.

Though I have some gripes that verge on the political for the limitations section I think this is an interesting and well-motivated work that deserves acceptance.

**Strengths:**

Detailed analysis and clear statement of technique

**Weaknesses:**

Yudkowsky citation. I think that stuff is fundamentally unserious and hurts my willingness to recommend strong accept or award as an actual NLP expert.

**Questions:**

Curious to see other reviews. My apologies for being severely time-constrained in writing my review.

**Limitations:**

In my opinion, perpetuating AI safety hype in academic papers is inappropriate, and citing Yudkowsky in particular is a negative signal. Opinions of others will vary and ultimately I don't think this is a reason to reject. Just wanted to register my discontent.

Otherwise discussion of limitations is strong.

---

> ### Author Rebuttal · Authors · 2023-08-09
>
> Thank you for your review! We are glad you liked our experimental setup, found our analysis detailed, and our results interesting.
>
> Regarding the functional decision theory citation, we agree that a reference to a peer-reviewed venue would be more appropriate. There is indeed a more appropriate citation for it [1] published in The Journal of Philosophy, one of the most prestigious venues of that field, which we'll cite as a primary reference instead. If you are interested we'd be happy to elaborate on functional decision theory and the role it plays in our arguments in the "Potential Implications" section.
>
> [1] Levinstein, Benjamin A., and Nate Soares. "Cheating death in damascus." The Journal of Philosophy 117.5 (2020): 237-266.

---

### Official Review · Reviewer_21mo · 2023-07-17

**Soundness:** 3 good
**Presentation:** 3 good
**Contribution:** 3 good
**Rating:** 6
**Confidence:** 3

**Summary:**

The authors introduce the phenomenon of internalization in LLMs, specifically "weak internalization" and "strong internalization" (out-of-context meta-leanring). Weak internalization refers to LLMs' improved performance on questions with consistent definitions rather than with inconsistent ones. Strong internalization involves LLMs' ability to provide better answers for variables with a defined tag representing a consistent definition, demonstrating out-of-context meta-learning. The paper includes ablations to support their findings and discusses the limitation of the work.

**Strengths:**

- The paper presents an interesting case of "internalization" in LLMs.
- Authors addressed the limitations of their work and also mention the lack of conclusive explanations for internalization in general.

**Weaknesses:**

Major Concerns.
- Not enough models are analyzed in ablations. The paper presents only the evaluation of Pythia and T5 family of models and claims that the internalization phenomenon is quite general. I would suggest performing experiments with more recent models such as LLaMa, T5Flan, etc.
- The number of datasets presented for evaluation is also quite small. Although, I understand that creating datasets for this specific format could be expensive.
- it is unclear if the size of a model affects the internalization phenomenon. Ablations of a few models varying their size would help to solve this doubt.
- Not clear how the phenomenon of internalization can be taken by the community in order to improve or avoid pitfalls regarding the development of LLMs

Minor comments:
- The paper mentions several times to look at the appendix but doesn't indicate to which section the reader should pay attention. I would suggest indicating the specific section in order to improve the readability of the manuscript.
- The notations of the datasets are quite difficult to follow, I think authors could provide a general overview (in a table or any other format) rather than explaining each component in line with the text. This would improve the readability of the paper.

**Questions:**

Take a look at the weakness presented in the previous section.

**Limitations:**

Yes, the authors addressed the limitations of their work and also mention the lack of conclusive explanations for internalization in general.

---

> ### Author Rebuttal · Authors · 2023-08-09
>
> Thank you for a thoughtful review and suggestions. We'll aim to address and resolve them below:
>
> > It is unclear if the size of a model affects the internalization phenomenon. Ablations of a few models [of] varying size would help [...].
>
> We agree that an ablation of the model size against the extent of internalization would be useful, so we ran this experiment on Pythia models up to 6.9b parameters; see the plot in the response PDF. We will include this plot in the paper and discuss it in Section 2.6, and add a similar plot for the T-REx dataset.
>
> >  I think authors could provide a general overview [of dataset notation] (in a table or any other format) [...]
>
> Thanks for the suggestion of adding an explanatory table for the dataset notation. We agree it would make it easier to understand what's going on, and will add one to the paper. We include a draft version of the table we'll incorporate into the paper in the attached PDF.
>
> > The paper presents only the evaluation of Pythia and T5 family of models [...]. I would suggest performing experiments with more recent models such as LLaMa, T5Flan, etc.
>
> Our paper already demonstrates internalization with a decoder-only transformer (both finetuned and trained from scratch), encoder-decoder transformer, and a convolutional network. Following your review, we ran our experiments on LLAMA2-7B, obtaining very similar results as with the Pythia model. In addition, we conducted our early experiments with the GPT-Neo model family; the results are also very similar to the Pythia ones. We will include these results in the appendix and reference them in the last paragraph of Section 2.6. We will be happy to include more models on top of these if you or other reviewers believe that'd be valuable. We think going beyond the models listed above is of somewhat limited value unless the model architectures are sufficiently different to expect the hypothetical results might differ.
>
> > The number of datasets presented for evaluation is also quite small. Although, I understand that creating datasets for this specific format could be expensive.
>
> We created and studied 4 datasets in the paper: 2 language-based, 1 based on an arithmetic task, and 1 computer vision. We believe constructing even more datasets would only be interesting if there is a particular setting that is not covered by these 4 in which there is a reasonable doubt the findings would be different. If you have suggestions of a setting where one could expect our findings might not generalize, please let us know!
>
> > Not clear how the phenomenon of internalization can be taken by the community in order to improve or avoid pitfalls regarding the development of LLMs
>
> Our primary goal with this work is to highlight what we believe to be an important phenomenon to the machine learning community. The main contribution of our paper is *establishing the existence and generality* of out-of-context meta-learning in the first place.
>
> We believe the phenomenon is of high relevance to the design and development of deep learning models. For one, internalization is clearly linked to generalization, and our experiments demonstrate that the model can learn to generalize in surprising ways. We also discuss important implications for AI Safety in Section 6. Studying this phenomenon further might reveal what mechanisms are responsible for internalization, yielding actionable recommendations for the development of deep learning models.
>
> We believe that much more work is needed to understand this phenomenon and its actionable implications -  we outline several directions for studying this phenomenon in section 4. However, we think this work 1) should be done by more groups than just ourselves, hence the value of putting a spotlight on the existence on the phenomenon 2) will require investigation beyond the scope of one paper.

---

> > ### Comment · Reviewer_gp6r · 2023-08-16
> > **"Not enough models..." as a critique is vapid, but the authors haven't addressed the "...to make a general claim" part**
> >
> > I definitely share the authors frustration with seeing the phrase "The paper presents only the evaluation of Pythia and T5 family of models [...]. I would suggest performing experiments with more recent models such as LLaMa, T5Flan, etc." in their review.
> >
> > After all, complaints in this vein often are a vapid and lazy criticism, as adding more models to the analysis renders the more surgical style of adding the tokens presented in this work more time consuming for questionable gain.
> >
> > **That being said, I do agree with the reviewer that *claims of general results should be soft* in light of the evaluation of only 2 classes of models, and would like to see the authors ensure this claim softness is present in their final paper should it be accepted.**
> >
> > **Curious to hear 21mo's thoughts on this matter in their response/updated review.**

---

> > > ### Author Response · Authors · 2023-08-21
> > >
> > > Thanks for the feedback gp6r! We agree that all the statements regarding the generality of the results should be adequately framed. Currently we are planning to make the following edits:
> > > - Line 55: change *Our results indicate that internalization is a general property* -> *might be a general property*
> > > - Line 546 (appendix C): change *To establish the generality of our results* -> *To investigate the generality of our results*
> > >
> > > Overall given that we reproduce out-of-context meta-learning with three kinds of decoder-only transformers (Pythia, GPT-Neo, LLAMA2), an encoder-decoder transformer (T5), and a ConvNet, and in both finetuning and training-from-scratch settings, **we think the main challenge to our results’ generality comes not from potential issues with other models, but from whether our toy datasets actually capture the relevant structure of the real-world data that LLMs are trained on.** We will add a sentence about this in the Discussion section.

---

> > > > ### Comment · Reviewer_21mo · 2023-08-21
> > > >
> > > > Thank you for your time and effort in addressing my major concerns. I disagree with the comment from reviewer gp6r. It's not a lazy critique; if the authors make a broad claim, they must prove it.
> > > >
> > > > After reviewing the authors' clarifications and their intention to moderate their claims, I decided to raise my score.

---

### Author Rebuttal · Authors · 2023-08-10

We would like to thank all reviewers for their constructive reviews, and are looking forward to further discussion. Summarizing the contents of the attached response PDF:

- A table explaining notation and data subset structure, as suggested by reviewers 1 and 3, and aimed at improving clarity as per reviewer 5.
- An ablation of the demonstrated phenomena across model sizes, as suggested by reviewer 1.
- Per-data-subset plots of train and validation losses accross our two-stage training procedure, to address a suggestion from reviewer 3.
- A plot for a new baseline experiment where definitions appear in the context of the questions, per suggestion of reviewer 3 and with aim to improve clarity as per reviewer 5.

We also ran several experiments and made a number of changes on top of the ones mentioned above, which we describe in our responses to individual reviewers.

---

### Decision · Program_Chairs · 2023-09-21

**Decision:**

Reject

**Comment:**

This work studies “internalization” in language models. A synthetic dataset is constructed whereby the “truthfulness” of information appearing in the LLM context can be manipulated. The dataset is used to investigate whether a fine-tuned language model relies on its training data prior and it is found that the model “internalizes” consistent-seeming definitions more than inconsistent-seeming ones.

On a surface level, I think this work studies an important topic. The reviewers are somewhat divided on the quality of this work so I took it upon myself to take a closer look. Upon further inspection, I think the work has some important issues, some of which are a sign of the times and some of which are specific to this paper:

* Relationship to prior work:

The concept of “internalization” is never formally defined. If I’d have to explain the findings to a colleague who is less familiar with recent literature, I would say something like “if the finetuning data is inconsistent with the pretraining data, the model performs worse” (weak internalization) and “if we further finetune the finetuned model, it will have learned that the token for inconsistent data means the data is inconsistent” (strong internalization).

To very simply describe the approach: take a pretrained model, finetune it on data that may or may not be consistent with the pretraining data (and demarcate this with a special token), then finetune it again on new data that uses these special tokens. The findings are unsurprising: the model relies on its pretraining prior (that’s why we pretrain) and finetuning leads the model to learn the meaning of the special tokens (that’s what happens when you train).

To go from that to “we are the first to describe this new important phenomenon that we call out-of-context meta learning” does a massive disservice to decades of machine learning research into the effect of priors, pretraining, learning theory and representation learning.  There has been a lot of work in e.g. domain generalization after pretraining, memorization vs generalization, multi-task learning, (catastrophic) forgetting, et cetera, all of which is relevant here.

Contrary to what the title suggests, the paper is not even about language models, because it shows the same phenomenon (again, unsurprisingly) holds on a simple MNIST task using a ConvNet. What follows is a worthwhile discussion of possible explanations for “internalization” but neither of them is thoroughly examined - I would recommend that the authors pursue these directions in more depth and try to properly explain the results. Lastly, the work tries to connect these findings to the AI safety literature, e.g. functional decision theory. The connection is a stretch, and frankly I have a hard time taking this work seriously as a result. Perhaps this work would be more suitable for a conference on AI safety that puts less emphasis on scientific rigor and is more amenable to unsubstantiated speculation.

* On "out of context":

The connection to meta-learning is taken for granted while it should be made explicit. I also have a hard time accepting the definition of “out of context” for what is observed here. I can see why it’s an interesting contrast with “in-context” learning (ICL), but the reason we needed a new word for ICL was exactly _because we already had a word for not-ICL_: we called it learning. If this work wants to coin a new term, it had better be very careful with its definitions and make very explicit what is different here.

* Other qualms:

I like the dataset design because it affords some control, but the work never goes beyond this toy setting and the MNIST example only adds to the confusion. Notationally, I’d advise using the logic symbols for true \top and false \bot instead of the different color scheme for Define. I think you should make the Define tokens proper randomly initialized special tokens and not a random sequence of characters. The phenomenon you are describing is called a counterfactual in logic, which could help you conceptually clarify what you are studying here.

In its current form, despite the relatively high average reviewer rating, I cannot recommend this work for acceptance. The topic is an interesting one and merits further — scientifically rigorous and properly grounded in the machine learning literature — study, so I would encourage the authors to dive deeper on these topics.

Reading through the reviews and the rebuttals, I think there is an emerging theme of confusion with the positioning of this work and its relationship to prior work. There is definitely merit in the research direction, but the work can be improved in terms of execution.